



# Heavy rainfall, floods, and flash floods in the context of solar wind coupling to the magnetosphere-ionosphere-atmosphere system

Paul Prikryl[1], Vojto Rušin[2], Emil A. Prikryl[3], Pavel Šťastný[4], Maroš Turňa[4], Martina Zeleňáková[5]

[1]Physics Department, University of New Brunswick, Fredericton, NB, E3B 5A3, Canada
[2]Astronomical Institute, Slovak Academy of Sciences, 059 60 Tatranská Lomnica, Slovakia
[3]Northern Ontario School of Medicine, Thunder Bay, ON, P7B 5E1, Canada
[4]Slovak Hydrometeorological Institute, 833 15 Bratislava, Slovakia
[5]Technical University of Košice, 040 00 Košice, Slovakia

*Correspondence to*: Paul Prikryl (paul.prikryl@unb.ca)

**Abstract.** Heavy rainfall events causing floods and flash floods are examined in the context of solar wind coupling to the magnetosphere-ionosphere-atmosphere system. The superposed epoch (SPE) analyses of solar wind variables have shown a tendency of severe weather to follow arrivals of high-speed streams from solar coronal holes (Prikryl et al., 2018). Precipitation datasets based on rain-gauge and satellite sensor measurements are used to examine the relationship between the solar wind high-speed streams and daily precipitation rates over several mid-latitude regions. The SPE analysis results show an increase in occurrence of high precipitation rates following arrivals of high-speed streams, including recurrence with a periodicity of 27 days. The cross-correlation analysis applied to the SPE averages of the green (Fe XIV, 530.3 nm) corona intensity observed by ground-based coronagraphs, solar wind parameters and daily precipitation rates show correlation peaks at lags spaced by solar rotation period. When the SPE analysis is limited to years around the solar minimum (2008-2009), which was dominated by recurrent coronal holes separated by ~120˚ in heliographic longitude, significant cross-correlation peaks are found at lags spaced by 9 days. These results are further demonstrated by cases of heavy rainfall, floods and flash floods in Europe, Japan, and the U.S., highlighting the role of solar wind coupling to the magnetosphere-ionosphere-atmosphere system in severe weather, mediated by aurorally excited atmospheric gravity waves.

## 1 Introduction

Extreme rainfall and flash floods have major societal and economic impacts, thus posing significant natural hazards (Gaume et al., 2009) that increasingly require careful flood risk assessment, mitigation strategies and recovery management (National Academies of Sciences, 2019). Although our knowledge and understanding of the multitude of processes that can result in heavy precipitation has advanced (Schumacher, 2019), prediction of extreme precipitation events continues to present difficult challenges in operational forecasting (Fritsch and Carbone, 2004; Villarini et al., 2010; Gourley et al., 2012; Schroeder et al., 2016). Recent climate change has caused an increase in extreme precipitation and flash floods (Groisman et al., 2005; Gutowski et al., 2008; Esposito et al., 2018, Schumacher, 2019). While there are many factors affecting heavy precipitation across a variety of atmospheric scales, the difficulty to predict flash floods stems partly from the fact that they often occur on small





spatial scales. Schumacher (2019) summarized some of the aspects of extreme precipitation and research areas aiming for improved understanding and prediction, which, among other things, will require interdisciplinary research collaborations in addition to basic research.

A possible link between solar magnetic sector structure and tropospheric vorticity was shown in the 1970s by Wilcox et al. (1973, 1974) who used the upper-level tropospheric vorticity area index (VAI), which is a proxy for extratropical storminess. This discovery prompted a search for a possible physical mechanism, such as the global electric circuit model and changes in relativistic electron flux (Tinsley, 2000; 2008), energetic solar proton events correlated with intensifications of cyclonic activity (Veretenenko and Thejll, 2004), which in turn would affect cloud microphysics (Tinsley, 2012). For further discussion and references, see, e.g., Prikryl et al. (2019).

Prikryl et al., (2009a) confirmed the "Wilcox effect" in both the northern and southern hemisphere winters. Severe weather, including severe winter storms, windstorms, snowstorms, and heavy rain events, and explosively developing extratropical cyclones have been examined by Prikryl et al. (2018) in the context of solar wind coupling to the magnetosphere-ionosphere-atmosphere system. They observed a tendency of significant weather events to follow arrivals of high-speed solar wind. The statistical results (Prikryl et al., 2009a; 2016; 2018) and a proposed physical mechanism (Prikryl et al., 2009b) were supported by cases of severe weather events, including two flash floods that occurred in Slovakia on July 16 and 24, 2001 (Prikryl et., 2018; their Fig. 10). Both flash floods closely followed arrivals of solar wind high-speed streams from coronal holes, and a series of convective supercells were associated with atmospheric gravity waves (AGWs) from sources in the high-latitude lower thermosphere that could have reached the mid latitude troposphere. The authors suggested that the AGWs played a role in triggering moist instabilities, thus initiating the convection.

In this paper we examine the occurrence of heavy rainfall events leading to floods and flash floods in the geophysical context of the solar wind coupling to the magnetosphere-ionosphere-atmosphere (MIA) system. The goal is to demonstrate the statistical link between the solar wind and the occurrence of heavy rainfall, and to identify the high-latitude sources of AGWs that may play a role in triggering the convection leading to heavy rainfall, floods, and flash floods.

## 2 Data sources

Prikryl et al. (2018) have described in detail the solar wind data obtained from the National Space Science Data Center (NSSDC) OMNIWeb http://omniweb.gsfc.nasa.gov (King and Papitashvili, 2005). The data include solar wind velocity, $V$, the interplanetary magnetic field (IMF) magnitude, $B$, the standard deviation, $\sigma_{Bz}$, of the z-component of the IMF, $B_z$, and proton density, $n_p$. These solar wind parameters are used to identify co-rotating interaction regions (CIRs) at the leading edge of high-speed streams from coronal holes, an interface between the fast and slow solar wind.






Measurements of the green coronal emission line (Fe XIV, 530.3 nm) by ground-based coronagraphs from 1939 to 2008 have been merged into a homogeneous coronal dataset (Rybanský, 1975; Rybanský et al., 2001, 2005) ftp://ftp.ngdc.noaa.gov/STP/SOLAR_DATA. The intensity of the emission line at 28.4 nm (Fe XV) observed by the SOHO Extreme ultraviolet Imaging Telescope (EIT) instrument was found to be closely correlated with intensity of the green corona

obtained by ground-based coronagraphs (Dorotovič et al., 2014). The coronal intensities are expressed in absolute coronal units (ACU) representing the intensity of the continuous spectrum from the center of the solar disk with a width of 1 Å at the same wavelength as the observational spectral line (1 ACU = 3.89 W m$^{-2}$ sr$^{-1}$ at 530.3 nm). The intensity depletions, called coronal holes, are sources of high-speed solar wind streams. While they are most prominent in polar regions (polar coronal holes) they can extend to low heliographic latitudes as depleted corona intensity and are observed at the limb, sometimes even

during total solar eclipses (Rušin et al., 2020). The green corona intensity synoptic charts at the solar central meridian are produced by averaging the intensities measured at the east and west limbs 14 days apart.

The radars of the Super Dual Auroral Radar Network (SuperDARN) (Chisham et al., 2007) map ionospheric convection in both the northern and southern hemispheres (vt.superdarn.org). Ground-based magnetometers from the Geophysical Institute Magnetometer Array (GIMA) (www.asf.alaska.edu/magnetometer/), International Monitor for Auroral Geomagnetic Effects

(IMAGE) (space.fmi.fi/image/), Geomagnetic Laboratory of the Natural Resources Canada (NRCan) (www.spaceweather.ca) and the Canadian Array for Realtime Investigations of Magnetic Activity (CARISMA) (www.carisma.ca/) are used to identify ionospheric currents as sources of AGWs. The magnetometer data were also accessed through SuperMAG (supermag.jhuapl.edu/mag) (Gjerloev, 2012) and INTERMAGNET (www.intermagnet.org).

De-trended GPS total electron content (TEC) maps are used to identify traveling ionospheric disturbances (TIDs) (Tsugawa et al., 2007). The Dense Regional And Worldwide INternational GNSS-TEC observation (DRAWING-TEC) project of the National Institute of Information and Communications Technology has developed the dense TEC mapping technique and shared the standardized GNSS-TEC data (https://aer-nc-web.nict.go.jp/GPS/DRAWING-TEC/).

Meteorological data were accessed online at web pages provided by various institutions including Environment Canada, Japan Meteorological Agency (JMA), Australian Government Bureau of Meteorology, Slovak Hydrometeorological Institute (SHMU), the University of Wisconsin-Madison Space Science and Engineering Center and the University of Washington, Department of Atmospheric Sciences. Copernicus (https://www.copernicus.eu/en), previously known as GMES (Global Monitoring for Environment and Security), is the European Programme for the establishment of a European capacity for Earth

Observation. It funds the project FloodList, a European system for monitoring and reporting floods and flooding news since 2013 (http://floodlist.com/). An earlier database of European flash flood data (Gaume et al., 2009) is provided at the Hydrate project webpage (www.Hydrate.tesaf.unipd.it). The rainfall data from SHMU stations across Slovakia and the SHMU annual flood reports (http://www.shmu.sk/sk/?page=128) provided data on significant rainfall leading to floods. The Tropical Rainfall



Measuring Mission (TRMM) Multi-Satellite Precipitation Analysis TMPA (3B42) Precipitation (version 7) using a 0.25˚ x

0.25˚ grid for latitudes ±50˚ was produced at the Goddard Earth Sciences Data and Information Services Center (GES DISC) (Huffman et al., 2016) (https://disc.gsfc.nasa.gov/datacollection/TRMM_3B42_Daily_7.html).

**2 Superposed epoch analysis of green corona intensity and solar wind variables keyed to onset of significant rainfall leading to floods and flash floods**


Prikryl et al. (2009a, 2009b, 2016, 2018, 2019) used the superposed epoch (SPE) analysis method to identify a tendency of severe weather, explosive extratropical cyclones and rapid intensification of tropical cyclones to follow arrivals of solar wind disturbances, including corotating interaction regions (CIRs) generated by solar wind high-speed streams (HSSs) from coronal holes and interplanetary coronal mass ejections (ICMEs). In the present paper we extend the analysis to examine the occurrence

of heavy rainfall leading to floods and flash floods in the geophysical context.

First, we use lists of flash floods in France from the Hydrate project database, flash floods in Slovakia (SHMU annual flood reports) and Poland (Ballesteros-Cánovas et al., 2015; their Table 4), and flash floods in Europe and USA archived by the FloodList project (http://floodlist.com/). Fig. 1 shows the results of the SPE analysis of time series of green corona intensity

and solar wind parameters keyed to start dates of flash floods in France (left panels), as well as in Slovakia on the northern slopes of the Tatra Mountains in Poland (right panels). In both cases, the results show an increase of the mean solar wind velocity $V$ from a minimum before the key day to a maximum after the key day, with peaks in $n_p$, $B$ and $\sigma_{Bz}$ about one day prior to the key time, which is a result of superposition of HSS/CIRs within the $n$ intervals centered on the flash flood day. This is further supported by the SPE analysis of green corona intensity showing depletions of the mean intensity at low heliographic

latitudes before the key time, which is the result of superposition of coronal holes, which are the sources of HSSs. These results indicate a tendency of flash floods to occur following the arrivals of HSS/CIRs from coronal holes.

Fig. 2 shows the SPE analysis results for flood events in Europe (Figs. 2a, b) and USA (Figs. 2c, d) with the key times defined by approximate starting dates of heavy rainfall obtained from the FloodList archive (2013-2019), excluding events caused by

tropical storms and rainfall persistent over many days. The results are similar to those in Fig. 1, showing patterns of increase in the mean solar wind velocity $V$ from minimum to maximum and peaks in $n_p$, $B$ and $\sigma_{Bz}$ near the key time. For each geographical sector, this reproducible pattern indicates a tendency of heavy rainfall/flood events to occur following arrivals of HSS/CIRs.

These results are further supported by a more comprehensive analysis using the SHMU precipitation database and the annual flood reports in Slovakia for a period of 2003-2019. Previously, Prikryl et al. (2018; their Figs. 7d, e, f) showed a tendency of significant weather events, including heavy rainfall, to follow arrivals of HSSs. Here we repeat the SPE analysis of solar wind



parameters keyed to significant rainfall events of at least 30 mm/24h recorded at 10 or more stations in 2003-2019 (Fig. 3). The results show a pattern indicating a tendency of these rainfall events to have occurred about 2 days after arrivals of HSSs

from coronal holes.

Heavy rainfall events used in Fig. 3 may or may not have resulted in floods. On the other hand, relatively moderate rainfall rates over a few days can also result in floods. To focus on floods, we compiled start dates of significant rainfall leading to floods as documented in the SHMU annual flood reports. The SPE analysis of the green corona intensity and solar wind

parameters keyed to these dates is shown in Figs. 4a and 4b, respectively. The histogram (Fig. 4c) shows a steep increase in the cumulative number of stations where the recorded 24-h rainfall exceeded a given threshold for each epoch day relative to the start of rainfall/flood events, regardless of the location of the SHMU stations. Of note, for some rivers, such as Myjava and Dunaj (Danube) the rainfall that contributed to floods occurred in Czech Republic, Austria and Germany but was not necessarily recorded by SHMU stations. While the available information on rainfall measurements outside Slovakia is used to

determine the key dates for some flood events in the SPE analysis (Figs. 4), the statistics of rainfall occurrence relative to the key date (Fig. 4c) only includes the SMHU stations. The SPE analysis of solar wind parameters (Fig. 4b) clearly indicates a tendency of the heavy rainfall/flood events to follow arrivals of solar wind HSS/CIRs, which is consistent with the results discussed above (Figs. 1 to 3).

## 4 High daily precipitation rates relative to arrivals of major HSS/CIRs

The above results of the SPE analysis of solar wind time series keyed by heavy rainfall leading to floods indicate a tendency of these events to follow arrivals of HSS/CIRs. Using the database of daily precipitation rates from rain gauges in Slovakia we now show more direct evidence to support these conclusions. Also, we use the TRMM satellite-based dataset of daily precipitation rates to provide further statistical evidence supporting these results.

To examine statistically the occurrence of significant rainfall in Slovakia relative to arrivals of HSS/CIRs the key time for the SPE analysis is now defined as the actual arrival time of major HSS/CIRs, and the total cumulative numbers of rain-gauge stations with recorded daily rainfall rate exceeding given thresholds are summed up for each epoch day.

For major HSSs reaching a maximum solar wind velocity of at least 600 km/s in the period of 2003-2019 (same as in Fig. 4),

Figs. 5a and 5b show the expected SPE analysis results for the green corona intensity and solar wind parameters, respectively. The superposition of HSS/CIRs relative to the well-defined interface between the fast and slow solar wind shows peaks in the mean $n_p$, $B$ and $\sigma_{Bz}$ near the key time while the mean solar wind velocity $V$ is rising from a minimum before to a maximum after the key time (Fig. 5b). In Fig. 5c, for each epoch day relative to HSS/CIR arrivals, the cumulative number of stations with the 24-h rainfall exceeding given thresholds are shown. The number of stations with significant rain increases from a

minimum before to a maximum after the HSS/CIR arrival. If more moderate HSS/CIRs (e.g., $V_{max} > 500$ km/s) are included,





the increase of significant rainfall occurrence following the key day 0 is still observed (Fig. 5d), although the relative increase is smaller. Conversely, for faster solar wind streams (Figs. 5e and 5f) significant rainfall occurrence show greater relative increases following the HSS/CIR arrivals. Independently of rain-gauge observations, the same relationship between HSSs and the satellite-based daily precipitation rates over Slovakia extracted from the TRMM dataset are shown in Figs. 5g, 5i and 5j.

This is consistent with the results discussed in Section 3 and confirms the tendency of increased occurrence of heavy rainfall following the HSS/CIR arrivals, which becomes more pronounced for stronger HSS/CIRs.

Coronal holes often persist for several solar rotations producing recurrent HSSs with a period of about 27 days. To show it, the SPE analysis is now extended to ±36 days from key time defined by arrivals of major HSSs reaching a maximum solar

wind velocity of at least 600 km/s (Fig. 6). The vertical dotted lines are shown for key time ± 27.28 days (called Carrington, or synodic rotation). The superposition of recurrent coronal holes, which include north and south coronal holes, results in depressions of the mean green corona intensity $I_{GC}$ (Fig. 6a) that are, in this case, centered at about zero heliographic latitude 3-4 epoch days before arrival times of recurrent HSSs. The white dotted line shows the mean $I_{GC}$ at zero latitude from the SPE analysis, with the mean over the period of 72 days subtracted. The standard error bar for one of the deviation minima, and the

ordinate scale bar corresponding to the color scale on the right are shown. Fig. 6b shows the superposition of solar wind plasma parameters that includes recurrent HSS/CIRs. As in Fig. 5c, but now for ±36 days from the key time, Fig. 6c shows cumulative numbers of rain-gauge stations in Slovakia with above-threshold daily precipitation rates, $N_{20mm}$, $N_{30mm}$, and $N_{40mm}$. The black line shows the precipitation rate averaged over all stations for each epoch day (smoothed by a 3-point running window). The histograms show not only the clear increase in precipitation after the key time that is already shown in Fig. 5c, but also similar

increases one solar rotation before and after the key time, i.e., an increase in precipitation following the arrivals of recurrent streams. We now compare variations among six variables ($V$, $I_{GC}$, $n_p$, $B$, $\sigma_{Bz}$ and the mean daily precipitation rate). The cross-correlation function (CCF) is computed for the mean solar wind velocity $V$ paired with the rest of the variables (Fig 6d; shown in colors corresponding to those in Fig. 6b, except for the green line representing $I_{GC}$). CCF ($I_{GC}$, $V$) has the maximum correlation coefficient at a lag of +1 day with secondary maxima at lags of −26 and +28 days, i.e., +1 day relative to the vertical

dotted lines representing recurrent streams. The relative lags for the main and secondary peaks of CCF ($B$, $V$) and ($\sigma_{Bz}$, $V$) are 1 or 2 days, and 2 or 3 days for ($n_p$, $V$). Most importantly, the peaks of the CCF (mean rate, $V$) are at lags of −27, 0 and +27 days, with correlation coefficients that are comparable with those for the other solar wind variables. These quantitative results further support the qualitative observations discussed above by confirming the tendency of an increase in precipitation following the arrivals of HSS/CIRs, including recurrence with a periodicity of 27 days. The latter is demonstrated once more

another way in Fig. 7. It shows the SPE analysis of green corona intensity, solar wind parameters, and cumulative number of rain-gauge stations in Slovakia with above-threshold daily precipitation rates, except the key times are lagged by ±27 days relative to the actual arrival times of HSS/CIRs used in Fig. 5, thus doubling the number of superposed time series. Expectedly, Figs. 7a and 7b show smaller amplitudes, because of averaging "imperfectly" recurrent streams (coronal holes and streams evolve over one solar rotation, shifting in longitude and arrival times, respectively), but the amplitude of the increase in the





numbers of stations Slovakia with high precipitation rates doubles, because they are summed up for two returns of recurrent
streams separated approximately by 54 days. If the SPE analysis is repeated for key times lagged by ±54 days, the results are
very similar, even though now the recurrent streams are separated by approximately 108 days (not shown).

Of course, HSS/CIRs and their sources, coronal holes, are more often spaced by less than 27 days, typically by ~9 or 13-14
days, depending on how many coronal holes cross the solar meridian in one solar rotation. Because their spacing varies over
a period of many years (Fig. 6), the SPE analysis averaging smears these intermediate variations. Recurrent coronal holes,
which are generally regularly spaced in heliographic longitude, are often present during the descending phase of the solar
cycle, and around solar minima. De Toma (2011) analyzed coronal holes during the extended minimum between Cycles 23
and 24. Two or three low-latitude coronal holes per solar rotation were observed by SOHO/EIT in 2008 separated by about
110°-120° in longitude (de Toma, 2011; see, their Figures 5 and 6). Recurrent HSSs and their impacts on the Earth can be best
viewed in a stack plot of time series ordered by solar rotation. Fig. 8 shows daily averages of solar wind velocity and the
numbers of stations in Slovakia with above-threshold daily precipitation rates, $N_{20mm}$, $N_{30mm}$, and $N_{40mm}$, for years 2008 and
2009 at the end of the solar cycle 23 that includes the solar minimum. Two recurrent major HSSs, and at times even a third
weaker one, were spaced by about 9 days on average and persisted for more than a year. They were later followed by two
streams spaced by about 14 days. The dashed lines in Fig. 8a indicate approximate arrivals of HSS/CIRs. Because the coronal
holes that produce the streams evolve over time, the arrival times of the HSS/CIRs can change from rotation to rotation, as
indicated by the tilted dashed line. Figs. 8b-d show time series $N_{20mm}$, $N_{30mm}$, and $N_{40mm}$ with approximate HSS/CIR times from
Fig. 8a also shown. On average, the occurrence of heavy rainfall followed the arrivals of major HSSs. This is very similar to
the effects on the ionosphere following major HSS/CIRs (Prikryl et al., 2012, their Fig. 7).


The SPE analysis for this period shows quite regularly spaced, predominantly southern coronal holes (Fig. 9a), and
corresponding recurrent streams, including CIRs (Fig. 9b). The mean $I_{GC}$ (white dotted line) is now extracted for 10°S
heliographic latitude. Fig. 9c shows the cumulative numbers of stations, $N_{20mm}$, $N_{30mm}$, and $N_{40mm}$, along with the 3-point running
average of the mean daily rate, all displaying variations that are very similar to those of solar wind parameters (Fig. 9b). Fig.
9d shows CCFs computed for $V$ paired with $I_{GC}$, $n_p$, $B$, $\sigma_{Bz}$ and the mean precipitation rate, all displaying primary peaks
separated by 27 days, and several secondary peaks spaced by 9 days. In particular, the CCFs $(B, V)$, $(\sigma_{Bz}, V)$, and (mean rate,
$V$) are almost identical, with maximum correlation coefficients at lag +1 day of 0.71, 0.78 and 0.71, respectively. The CCF
(mean rate, $B$) (not shown) peaks at lag zero with a maximum correlation coefficient of 0.55. Of note, $\sigma_{Bz}$ is a measure of solar
wind Alfvén wave amplitudes that are largest following arrivals of HSS/CIRs but continue to be relatively high inside the
HSS, while $B$ and $n_p$ tail off faster (Fig. 9b), which is likely the reason why the peak of the CCF $(\sigma_{Bz}, V)$ cross-correlation
coefficient is highest.



These results are similar to the "Wilcox effect" (Prikryl et al., 2009a; their Figures 3 and 4), which is mentioned in the Introduction. The HSSs from coronal holes are anchored in the large-scale solar magnetic field structure that is extended into

the interplanetary space by solar wind. The magnetic sector boundaries, known as the heliospheric current sheet (HCS) (Smith et al., 1978; Hoeksema et al., 1983), often precede the HSS interfaces by about one day, unless the two coincide. If the key times in the above SPE analysis are defined by the magnetic sector boundary crossing (SBC), instead of HSS/CIRs, results that are similar to those in Fig. 9 are obtained (not shown).

**5 Cases of extreme rainfall and flash floods in Slovakia**

In the Sections 3 and 4 we showed statistical evidence of the relationship between the solar wind HSS/CIRs and the occurrence of significant rainfall, focusing on the period 2003-2019 covered by the SMHU annual reports. We now discuss specific cases of extreme rainfall and flash floods in Slovakia, where the most affected region by floods is north-eastern Slovakia. Among

the most vulnerable areas in recent years are the river basins of the Topľa, Ondava, Hornád, Torysa, Laborec and Poprad rivers. Under the influence of long-lasting and intense precipitation, numerous flood waves arise, which may not be adequately mitigated by manipulating hydraulic structures. The water levels in many profiles on watercourses reached historical maxima in the above-mentioned regions. Regular occurrence of floods in eastern Slovakia is given primarily by a total land water regime, which is dominated by the very low absorptive power of heavy clayey soils of the Flysch zone (Azañón et al., 2010),

and adverse conditions of forests in eastern Slovakia.

One locality that has been repeatedly affected by flash floods in Slovakia is the High Tatras mountain range (Tatra National Park). In general, mountainous regions are often associated with flash floods, likely because of orographic enhancement of precipitation and anchoring of convective events, as well as various aspects of topographic relief that promote rapid

concentration of streamflow (Borga et al., 2011). The Tatra Mountains are the highest part of the Carpathians and this alpine massif in northern Slovakia has the highest total annual precipitation. Similarly, the northern slopes of the Tatra Mountains in Poland are affected by flash floods that are included in the SPE analysis in Fig. 1.

The most extreme flash flood that affected the southern slopes of the massif occurred on June 28-29, 1958 (Pekárová et al.,

2011). Several stations recorded daily precipitation rates exceeding 50 mm with the maximum reaching 170 mm at Skalnaté pleso (1778 m a.s.l.) (Šamaj et al., 1985). Although there were no spacecraft monitoring the solar wind at that time, coronal holes, the sources of HSSs, can be identified in synoptic maps of green corona intensity observed at high-altitude observatories. Furthermore, sector boundary crossings can be estimated from ground-based magnetograms (Svalgaard, 1975). Fig. 10a shows the synoptic map of green corona intensity indicating large southern coronal holes during this period. Here we are concerned

with the coronal hole prior to June 27, a source of HSS that impacted the Earth around June 28/29 (SBC is shown by an asterisk in Fig. 10b). There is also a possibility of an ICME from the bright coronal region on the Sun that may have arrived at about



the same time, thus amplifying the intensity of the impact on the Earth's magnetosphere and making it more geo-effective. The 3-hourly *Kp* index of geomagnetic activity (https://www.swpc.noaa.gov/products/planetary-k-index) reached very high values (Fig. 10c) during the strong geomagnetic storm with *Dst* index (Gonzalez et al., 1994) reaching −180 nT early on June 29, which coincided with the heaviest rainfall.

Fig. 11 shows another case of a severe flash flood that occurred on June 29-30, 1973. It followed the arrival of an exceptionally fast HSS (> 750 km/s; Fig. 11b) from a large northern coronal hole (Fig. 11a). The asterisks show the stream interfaces of two CIRs, both causing a significant increase in geomagnetic activity (Fig. 11d) and triggering moderate geomagnetic storms on June 23/24 and 28/29. Both geomagnetic storms coincided with heavy rainfall events. The second one, with a maximum daily rate of 102.5 mm at Skalnaté pleso (1778 m a.s.l.) on June 30, caused a flash flood in the High Tatras.

In July 2008, the maximum precipitation (144.6 mm) was observed in Tatranská Javorina on June 23, and the total between July 20 and 24 reached 260.2 mm, the largest recorded at this station. This was a year of solar minimum, and the green corona intensity was very low. However, a coronal hole can be identified (Fig. 12a), and a small compact coronal hole was observed by SOHO in the EUV images. The arrival of a strong HSS reaching a maximum *V* of 650 km/s, and a broad/double CIR with two stream interfaces on July 21 and 22 (the first of them marked with asterisk in Fig. 12b) was followed by moderate geomagnetic activity (Fig. 13d). Two spikes of intense rainfall were recorded at many stations.

The last major flood in Tatra National Park was caused by heavy rainfall from July 17 to 18 (Fig. 13c) with the precipitation maximum of 123 mm on July 18 that closely followed the arrival of HSS/CIRs on July 17 (Fig. 13b) from a large and structured north-south coronal hole (Fig. 13a). While the planetary *Kp*-index was relatively low (Fig. 13d), the strong CIR on July 16/17 produced by the stream that initially reached $V_{max} \approx 440$ km/s but a few days later exceeded 500 km/s, triggered intense auroral substorm activity observed by the IMAGE magnetometers on July 16 (https://space.fmi.fi/image/). The CIR was followed by large amplitude ultralow frequency (ULF) fluctuations of the ground magnetic field caused by ionospheric current fluctuations early on July 17. These are the sources of AGWs discussed in the next section.

In this section we discussed only some of the most severe flash floods on the southern slopes of the High Tatras mountain range in Slovakia. Two flash floods in July 2001 were already mentioned in the Introduction and have been discussed elsewhere (Prikryl et al., 2018). Flash floods on the Polish side of this mountain range and flash floods mentioned in the SMHU annual reports (2003-2019) also showed a tendency to follow arrivals of HSS/CIRs (Figs. 1c and 1d).

**6 Cases of heavy rainfall leading to floods and flash floods in Europe, Japan, and the United States**





We now discuss cases of heavy rainfall, floods, and flash floods in the geophysical context of solar wind MIA coupling. The hourly solar wind OMNI data ($V$, $B$, $n_p$ and $\sigma_{Bz}$) show a series of major HSS/CIRs (red asterisks) and ICMEs (orange triangle) in July 2017 and June-July 2018. The negative deflections of the *Dst* index (green line) show geomagnetic storms that were triggered by HSS/CIRs, the most intense one in conjunction with an ICME on July 16. The black symbols mark the start days/times of heavy rainfall that caused floods in Europe (triangles), Japan (circles) and the USA (squares). Most of these

events closely followed the arrivals of HSS/CIRs or ICMEs. Because the actual start times of heavy rainfall events were generally not available, most of the symbols are shown at 12:00 UT, except for cases when the actual start time is known.

The orange symbols (◊) in Fig. 14 mark start days/times of significant rainfall that caused floods in Slovakia (taken from the annual reports on floods). The orange dots at the top show daily precipitation rates measured at SHMU stations. The floods on

the rivers Bodrog and Ondava that were caused by significant rainfall (July 1 and 10), closely followed arrivals of major HSS/CIRs. The floods on the Hron river tributaries (July 24) occurred in the peak of a strong and structured HSS (Fig. 14a) from a wide and structured coronal hole.

In addition to floods and flash floods, Figs. 14 shows daily precipitation rates measured at SHMU stations in Slovakia (orange

dots). Also shown (in purple), are maximum daily rates and number of grid cells with rainfall exceeding 30 mm, from the Deutscher Wetterdienst (DWD) REGNIE dataset. These variables show significant increases following most of the HSS/CIRs.

Fig. 15a shows the synoptic map of green corona intensity starting with a large southern coronal hole at the end of June, followed by a northern structured coronal hole extended at low heliographic latitudes between July 5 and 10. They were sources

of two major HSSs that impacted the Earth with CIRs arrivals on July 1 and 9, and a minor HSS/CIR on July 6 (Fig. 15b), each causing an increase in geomagnetic activity (Fig. 15d). The arrivals of the two major HSS/CIRs were associated with maxima in daily precipitation, on June 1 and 10, the latter exceeding 60 mm (Fig. 1c).

The last HSS at the end of July 2017 is presented in Figs. 15e-h. Fig. 15e shows two large north-south coronal holes joined into a wide structured coronal hole in southern latitudes that was a source of a very strong and structured HSS (Fig. 15f). The

arrival of this HSS/CIR on July 20 was followed by heavy precipitation in Slovakia that peaked on July 23 exceeding 80 mm, followed by more rain, and floods the next day (Fig. 15g). Furthermore, a flash flood caused by extreme rainfall exceeding 230 mm occurred in Kansas City on July 22, and heavy rainfall exceeding 300 mm in many places in Japan was recorded on July 22-24 (Fig. 14a).

Focusing on floods in Slovakia, a selection of the Meteosat RGB Composites Airmass images (composite based upon data from IR and WV channels) in Fig. 16 show intensifying mesoscale systems on July 1, 10 and 24. Some of the convective cells/bands over Slovakia caused heavy rainfall leading to floods. On July 1, starting at ~02:00 UT a series of cloud/rain bands





called striated delta cloud systems (Feren, 1995) developed in the warm frontal zone of an intensifying extratropical cyclone passing over eastern Slovakia and Poland (Fig. 16a).


We now present relevant geophysical data collected by the SuperDARN radar in Halkasalmi, Finland, with beams pointing poleward over Svalbard measuring line-of-sight velocity $V_{los}$ of convection/flows in the ionosphere (Chisham et al., 2007). After 20:00 UT on June 30, the radar observed a series of pulsed ionospheric flows (PIFs) with periods from 10 to 20 min between N70° and 75° latitudes (between the northern tip of Scandinavia and Svalbard). Fig. 17 shows the median filtered

line-of-sight velocity $V_{los}$ for radar beam 11 as a function of geographic latitude. Fig. 18a shows a time series of $V_{los}$ for range gate 22 (75° latitude). The Fast Fourier Transform (FFT) power spectrum of the detrended time series (dotted line) shows a peak at 1.1 mHz (~15 min). PIFs are sources of atmospheric gravity waves (Prikryl et al., 2005) that were likely launched several hours before the striated delta clouds developed on July 1.

Before we discuss AGWs, their sources in the high-latitude lower thermosphere, and their propagation upward and downward, it should be noted that when the AGW group velocity is downward, the phase velocity is upward, and vice versa, with the two velocity vectors being perpendicular to each other. For clarity, we will refer to downward/upward group (wave energy) propagation as down/up-going AGWs to distinguish from upward/downward AGW phase propagation.

The equatorward propagating down-going AGWs launched by PIFs (Fig. 17) would reach the troposphere. If ducted, similarly to the case shown in Prikryl et al. (2018; their Figure 15), down-going AGWs could have reached the unstable warm frontal zone of the cyclone over Slovakia and initiated/triggered slantwise convection resulting in a series of cloud bands (Fig. 16a). Of note, the number of cloud bands in this striated delta cloud was about the same as the number of PIFs (Figs. 17a and 18a). A tendency of striated delta clouds to develop following the arrivals of HSS/CIR has been shown previously (Prikryl et al.,

2018, their Fig. 9), and the present case is consistent with these statistical results and the case study presented there.

The increased divergence between 150-300 mb in the warm frontal zone associated with the striated delta cloud (Fig. 16a) suggests an unstable region, and the strong mid-upper level winds with the northward component indicate winds opposing the direction of the incoming/down-going AGWs. These conditions are conducive to over-reflection of AGWs, with a possibility

of amplification (Jones, 1968; Cowling et al., 1971; McKenzie, 1972; Eltayeb and McKenzie, 1975).

The warm frontal zone is known for moist symmetric instability (MSI) (Schultz and Schumacher, 1999), where a slantwise convection can be readily initiated by even small (infinitesimal) displacements of moist air. It was suggested that MSI can be triggered by over-reflecting down-going AGWs that originate in the lower thermosphere at high (auroral) latitudes. Previously,

cloud/rain bands were linked to possible sources of AGWs or observed traveling ionospheric disturbances (TIDs) in a few cases (Prikryl et al., 2009b; Prikryl et al., 2018, including their supplementary material).





On July 10, a string of convective cells developed in the cold frontal zone passing over Slovakia and Poland (Fig. 16b), one of
them causing heavy rainfall (Fig. 15c) and floods (Fig. 14a) in Slovakia. We note the similarity with the heavy rainfall/flash
flood event that involved a string of supercells (Prikryl et al., 2018, their Fig. 14) that were likely triggered by down-going
aurorally generated AGWs reaching the troposphere, while the up-going AGWs were observed in the ionosphere.
Unfortunately, this time, strong HF absorption wiped out most of the ionospheric radar backscatter and no PIFs could be
observed at high latitudes. Also, only weak radar ground scatter was observed indicating TIDs between 12:00 and 20:00 UT
(not shown). However, AGWs were likely launched by pulsing ionospheric currents observed over Svalbard on July 10, several
hours before the string of convective cells developed. Fig. 18b shows the magnetic field X component observed by the IMAGE
magnetometer NAL. The FFT spectrum of the detrended time series shows a peak at 0.4 mHz (~40 min). High divergence at
the mid-upper level in the cold frontal zone indicates an unstable region where the convection occurred (Fig. 16b). Down-
going AGWs over-reflecting in the unstable region could have triggered/initiated convection.

Fig. 16c shows a similar instance of convective cells triggered over Hungary that subsequently moved over Slovakia and
caused heavy rainfall and flood on July 24 (Figs. 14 and 15g). Again, PIFs could not be observed by the radar but pulses of
ionospheric currents were observed over Svalbard several hours before these convective cells were initiated. Fig. 18c shows
the magnetic field X component pulsation observed by the IMAGE magnetometer NAL. The FFT spectrum of the detrended
time series shows two prominent peaks at 0.4 mHz (~40 min) and 0.75 mHz (~20 min). These ionospheric currents likely
launched down-going AGWs that could have triggered the convection cells in the unstable region of high divergence and
strong north-eastward winds at the mid-upper level (Fig. 16c).

Fig. 16d shows a string of convective cells in the intensifying mesoscale system Medusa that produced heavy rainfall/flash
floods in Greece on July 16, 2017 (Fig. 14a). The overlay CIMSS data products are missing for this day but the next day still
shows strong divergence and southerly winds at mid-upper level over Greece (Fig. 16d). Moist instabilities could have been
released by AGWs launched by strong ionospheric current pulses that were observed over Svalbard several hours before
convection developed on July 16. Fig. 18d shows large-amplitude pulsation of the ground magnetic field X component
observed by the IMAGE magnetometer NAL. The FFT spectrum of the detrended time series shows a peak at 0.55 mHz (~30
min). Weak traces of TIDs caused by the up-going AGWs in the thermosphere were observed in the ground scatter by the
Hankasalmi radar beam 13 (not shown).

The strings of convective cells in Fig. 16 are similar to certain types of mesoscale convective lines (Bluestein and Jain, 1985;
their Figure 1). These authors identified four distinct types of mesoscale convective lines of cells that form squall lines, namely,
"broken line", "back building", "broken areal", and "embedded areal" types. The squall lines form in a conditionally and
convectively unstable atmosphere. The "broken line" forms typically along a cold front with multi-cells appearing at about the





same time and transforming "into a solid line as the area of each existing cell expands and new cells develop" (Bluestein and Jain, 1985). This type could represent the cases shown in Figs. 16b and 16d. The "back building" type "consists of the periodic appearance of a new cell upstream, relative to cell motion" and can form along different types of surface boundaries. This type pertains to the case in Fig. 16c, and the cases discussed by Prikryl et. al, 2018, where new cells appeared periodically. The

other squall line types may include the warm frontal bands and wide cold frontal bands (for detailed description of their characteristics, see, Bluestein and Jain, 1985). Similarly to striated delta clouds (Prikryl et al., 2018; their Fig. 9), the SPE analysis of solar wind data keyed to dates from the list of cases (Bluestein and Jain, 1985; their Table 1) shows a tendency of these four types of squall lines to follow arrivals of HSS/CIRs/ICMEs (not shown).

The HSS and a broad CIR, combined with an ICME on July 16 (Fig. 14a), caused a moderate geomagnetic storm (*Dst* = −72 nT) starting with the arrival of the interplanetary shock at about 06:00 UT. Fig. 19 shows strong magnetic field perturbations (ionospheric currents) observed by the IMAGE array over a range of latitudes from Fennoscandia to Svalbard. The ionospheric current pulses launched large-amplitude AGWs that produced large- and medium-scale TIDs observed in maps of the GPS Total Electron Content (TEC) over much of Europe as well as the North America on July 15 and 16 (Fig. 20) ([https://aer-nc-](https://aer-nc-)

[web.nict.go.jp/GPS/EUROPE/MAP/2017/197/index.html](https://web.nict.go.jp/GPS/EUROPE/MAP/2017/197/index.html)). During this time, in addition to the flash floods in Greece, already discussed above, intensified mesoscale systems produced heavy rainfall and flash floods in Turkey and southern England. These events are marked by triangles in Fig. 14a. In the U.S., deadly flash floods struck in Arizona with thunderstorms and heavy rain affecting areas around Payson between 21:00 and 23:00 UT on 15 July: "The National Weather Service issued a flash flood warning for Gila County and estimated that up to 38 mm of rain fell over the area in an hour… A further storm on

16 July caused flooding in the Phoenix area. Litchfield Park recorded 52 mm in 24 hours 16 to 17 July, 2017" ([http://floodlist.com/america/usa/flash-floods-arizona-july-2017](http://floodlist.com/america/usa/flash-floods-arizona-july-2017)). The observation here is that large-amplitude TIDs (Fig. 20) were caused by up-going AGWs from sources in the high-latitude lower thermosphere. In turn, the down-going AGWs that can reach the troposphere could have triggered/released moist instabilities initiating convection.

Finally, we briefly discuss the second period in June-July 2018 (Fig. 14b) that shows a series of 6 major HSS/CIRs. On June 25, an ICME (orange triangle) triggered a moderate geomagnetic storm that was further intensified with the arrival of a strong HSS/CIR (June 26; red asterisk). While this HSS was weakening, another ICME caused a small solar wind disturbance on June 30. Subsequent HSSs produced broad CIRs with large density and magnetic field fluctuations that resulted in strong MIA coupling.


In this period, heavy rainfall events caused large number of floods and flash floods in the U.S., Europe, and Japan. All of them followed the arrivals of HSS/CIRs or ICMEs, and some of them occurred on different continents at about the same time. On June 26-29, heavy rain caused floods in Greece, Bulgaria, Romania, and Slovakia. Following the ICME on June 30, extreme rainfall that occurred in Iowa (24-hour rate of 221 mm in Ankeny and 213 mm in Johnston) on June 30 to July 1 caused deadly





flash floods. The next major HSS/CIR that arrived on July 5 was preceded by a minor but broad HSS/CIR on July 3-4. Flash
       floods occurred in Italy and the U. S. (Minnesota, Houston, and Texas), and on June 5, heavy rain exceeding 500 mm in 24
       hours caused flash floods and landslides in Japan. Following the HSS/CIR on July 17-18, severe floods and flash floods
       occurred in the U.S. (Washington DC and Massachusetts) and Slovakia (http://floodlist.com/europe/slovakia-floods-tatra-
       mountains-july-2018) (Fig. 13). On July 21-22, in the peak of a structured HSS, flash floods occurred on the upper portion of

river Váh in Slovakia.

       Furthermore, daily precipitation rates measured at SHMU stations in Slovakia, as well as the maximum daily rates and the
       number of REGNIE grid cells with rainfall exceeding 30 mm in Germany (Fig. 14b), show increases following the HSS/CIRs
       including the HSS/CIR on July 10. The European Severe Weather Database (https://eswd.eu/cgi-bin/eswd.cgi) includes

confirmed reports of heavy rain causing floods in several countries from July 10-12, including a large area of Poland that was
       affected by heavy rain and floods on July 11. Of course, the information about flood events used in this paper is not complete
       and more data, including the NOAA Storm Events Database events in the U.S. (https://www.ncdc.noaa.gov/stormevents/) will
       be examined in the future.

**7. Discussion and Summary**

       We have examined heavy rainfall leading to floods and flash floods in the context of solar wind, linking the occurrence of such
       weather events to arrivals of solar wind HSSs, which are dominated by Alfvén waves (Belcher and Davis, 1971; Tsurutani and
       Gonzalez, 1987). Solar wind MIA coupling generates medium- to large-scale AGWs globally propagating from sources in the

lower thermosphere at high latitudes both upward and downward (Mayr et al., 1984a, 1984b, 1990, 2013; Hocke and Schlegel,
       1996). The coupling is most intense when the HSS/CIRs and/or interplanetary shocks at the leading edges of ICMEs arrive
       and generate large-amplitude AGWs originating in the lower thermosphere at high latitudes. Pioneer spacecraft in 1970s were
       the first to observe interaction regions between adjacent solar wind streams (Smith and Wolfe, 1976) and their geo-
       effectiveness has been well established (Tsurutani et al., 2006). Solar wind magneto-hydrodynamic waves, including Alfvén

waves, modulate the high-latitude ionospheric currents that can generate globally propagating AGWs (Prikryl et al., 2005; see,
       references therein).
       Aurorally excited AGWs can reach the troposphere and can be ducted in the lower atmosphere over long distances, thus
       reaching low latitudes. At the reflection point in the troposphere they can trigger moist instabilities to release latent heat, which
       in turn leads to intensification of extratropical cyclones (Prikryl et al., 2009b; 2016; 2018) and tropical cyclones (Prikryl et al.,

2019). These studies showed that explosive development of extratropical cyclones and rapid intensification of tropical cyclones
       tend to follow arrivals of solar wind HSSs and/or ICMEs.
       Prikryl et al. (2009b) suggested that in the extratropical cyclone warm frontal zone, with warm air advection over the cool air
       mass ahead, the over-reflecting gravity-wave-induced vertical lift may trigger/release moist symmetric instability (MSI) at



near-threshold conditions and thus initiate slantwise convection. Latent heat release associated with the mesoscale slantwise convection has been linked to explosive cyclogenesis (e.g. Kuo and Low-Nam, 1990). Callaghan and Power (2016) described conditions, including vertical wind structure, that lead to extreme rainfall and major flooding in southeast Australia. They found that most cases exhibited anticyclonic turning wind-direction with increasing height and warm moist air advection. They observed "typically east northeasterly winds at 850 hPa turning anti-cyclonically with height to northerly winds by 500 hPa." The downward propagating AGWs, in this case from the lower thermosphere sources at high southern latitudes, may encounter opposing winds in the upper troposphere, a condition conducive to over-reflection, that could release the MSI and trigger convection. Major floods in southeast Australia (Callaghan and Power, 2016) appear to have a tendency to occur following HSS/CIRs with results similar to Figs. 1 and 2, which will be discussed in a future publication.

Prikryl et al. (2018) discussed two flash floods in Slovakia that closely followed arrivals of two major HSS/CIRs in July 2001. They identified pulsing ionospheric currents as a source of AGWs that were observed in the ionosphere as TIDs. A few hours later a series of convective cells that formed in sequence were observed in infrared satellite images. These observations suggested that down-going AGWs played a role in triggering instabilities and initiating convection in the troposphere. One of the supercells caused heavy rainfall and flash floods in Slovakia. Cases of striated delta clouds were also linked to possible auroral sources of AGWs (Prikryl et al., 2018; supplementary material).

Consistent with these previously published results, the statistical results presented in Section 3 show that heavy rainfall events leading to floods and flash floods tend to follow arrivals of HSS/CIRs. The SPE analysis of green corona intensity and solar wind parameters keyed to the onset of heavy rainfall taken from several databases/lists of events in Europe and USA (Figs. 1 and 2) reproducibly show patterns that indicate this tendency, as discussed in Section 3. These conclusions are supported by the analysis of more comprehensive databases of daily precipitation rates and annual flood reports in Slovakia for a period of 2003-2019 using a twofold approach. First, the SPE analysis of green corona intensity and solar wind parameters is keyed to significant rainfall events of at least 30 mm/24h recorded at 10 or more stations (Fig. 3). Focusing on floods in Slovakia, the SPE analysis of the green corona intensity and solar wind parameters keyed to dates/times of significant rainfall leading to floods clearly indicates a tendency of these flood events to follow arrivals of solar wind HSS/CIRs (Fig. 4). Second, with the key times defined by the arrival times of major HSS/CIRs (Section 4), the SPE analysis (Fig. 5) shows an increase in the cumulative number of stations that measured daily precipitation rates above given thresholds. Such a relationship between solar wind HSSs and daily precipitation rates is also found using the satellite-based daily precipitation dataset TRMM. Finally, extending the SPE analysis to ±36 days about the key times defined by arrivals of major HSS/CIR reveals a similar response at epoch days ±27 due to recurrent streams. The cross-correlation between the SPE averages of the green corona intensity as well as solar wind parameters and mean daily precipitation rates quantitatively confirm the observed tendency of high precipitation occurrence following the HSS/CIRs.





In Sections 5 and 6, the tendency of heavy rainfall and floods following arrivals of HSS/CIRs is shown in case studies (Figs. 10-15), some supported by satellite images (Fig. 16) of intensifying mesoscale systems displaying striated delta clouds or strings of convective cells. Pulsed ionospheric currents (Figs. 17 to 19) are sources of AGWs that can be observed in the ionosphere (Fig. 20). Ray tracing of gravity waves (Prikryl et al., 2005; 2019) and simulations of gravity wave propagation using the Transfer Function Model (Mayr et al., 1984a; 2013; Prikryl et al., 2016; 2018) showed that AGWs can reach the troposphere and play a role in triggering moist instabilities and initiating convection leading to heavy rainfall and floods.

Recently, Hagiwara and Tanaka (2020) performed theoretical analysis of propagation of AGWs in the lower atmosphere using an expansion in three-dimensional normal mode functions. They showed that the waves can propagate downward to the troposphere as attenuating gravity waves, with the amplitude reduced by a factor of $10^3$ to $10^4$ in the troposphere. They found that "the wave propagations and reflections at the surface create an anti-node of geopotential at the bottom of the atmosphere corresponding to the vertical width of the initial state of the impact. On the other hand, standing waves in temperature create a node at the ground surface." They suggested that "due to the standing waves generated in the lower troposphere, the atmospheric stability is altered by the passage of the gravity waves in the meridional direction," and that the change in the stability parameters can affect the development of cyclones. Further theoretical and computational research examining the dynamics of down-going AGWs and their influence in the troposphere is needed to complement the data-driven analysis of this phenomenon.

## 8. Conclusions

Heavy rainfall causing floods and flash floods tends to follow arrivals of solar wind high-speed streams from coronal holes. The superposed epoch (SPE) analysis results show an increase in precipitation rates following arrivals of high-speed streams, including recurrence with a periodicity of 27 days. The cross-correlation analysis applied to the SPE averages of green corona intensity, solar wind parameters and daily precipitation rates show correlation peaks at lags spaced by solar rotation period. When the SPE analysis is limited to years around solar minimum (2008-2009), correlation peaks at lags spaced by 9 days are also revealed, which is a result of high-speed streams from coronal holes spaced in heliographic longitude by approximately 120˚. These quantitative results confirm the tendency of an increase in precipitation following the arrivals of high-speed streams, which is further demonstrated by cases of heavy rainfall, floods and flash floods in Europe, Japan, and the U.S. The role of aurorally generated atmospheric gravity waves as the mechanism mediating the influence of the solar wind-magnetosphere-ionosphere-atmosphere coupling on the troposphere is suggested. Down-going gravity waves from sources in the lower thermosphere can over-reflect in the upper troposphere and trigger/release existing moist instabilities, initiating convection and latent heat release, the energy leading to intensification of storms.



## Acknowledgements

This research was supported by the University of New Brunswick and the Public Safety Geosciences program of the Natural Resources Canada, Earth Sciences Sector, and by the VEGA project 2/0048/20 (Slovak Academy of Sciences). The rain-gauge data were provided by the Slovak Hydrometeorological Institute (SHMU). The SHMU with contribution by Slovak Water Management Enterprise (SWME) publish annual flood reports (in Slovak) that provide data on floods and significant rainfall leading to the floods in Slovakia since 2003. Also, this publication benefited from the project "Scientific support of climate change adaptation in agriculture and mitigation of soil degradation" (ITMS2014+ 313011W580) supported by the Integrated Infrastructure Operational Programme funded by the European Regional Development Fund (ERDF). Contributions by the ACE, Geotail, IMP-8, SoHO and Wind spacecraft teams, the NSSDC OMNIWeb, and SuperDARN and IMAGE projects are acknowledged. SuperDARN is a collection of radars funded by national scientific funding agencies of Australia, Canada, China, France, Italy, Japan, Norway, South Africa, United Kingdom, and the United States of America. Meteorological data were accessed online at web pages provided by various institutions including Environment Canada, Japan Meteorological Agency (JMA), Australian Government Bureau of Meteorology, Slovak Hydrometeorological Institute (SHMU), the University of Wisconsin-Madison Space Science and Engineering Center and the University of Washington, Department of Atmospheric Sciences. Copernicus, previously known as GMES (Global Monitoring for Environment and Security), is the European Programme for the establishment of a European capacity for Earth Observation. It funds a project FloodList, the European system for Earth monitoring and reporting floods and flooding news since 2008. An earlier database of European flash floods was collected by the HYDRATE project. Daily precipitation dataset REGNIE is freely available from the Deutscher Wetterdienst (DWD) Climate Data Center (CDC). The Tropical Rainfall Measuring Mission (TRMM) Multi-Satellite Precipitation Analysis TMPA (3B42) Precipitation (version 7) was produced at Goddard Earth Sciences Data and Information Services Center (GES DISC).

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

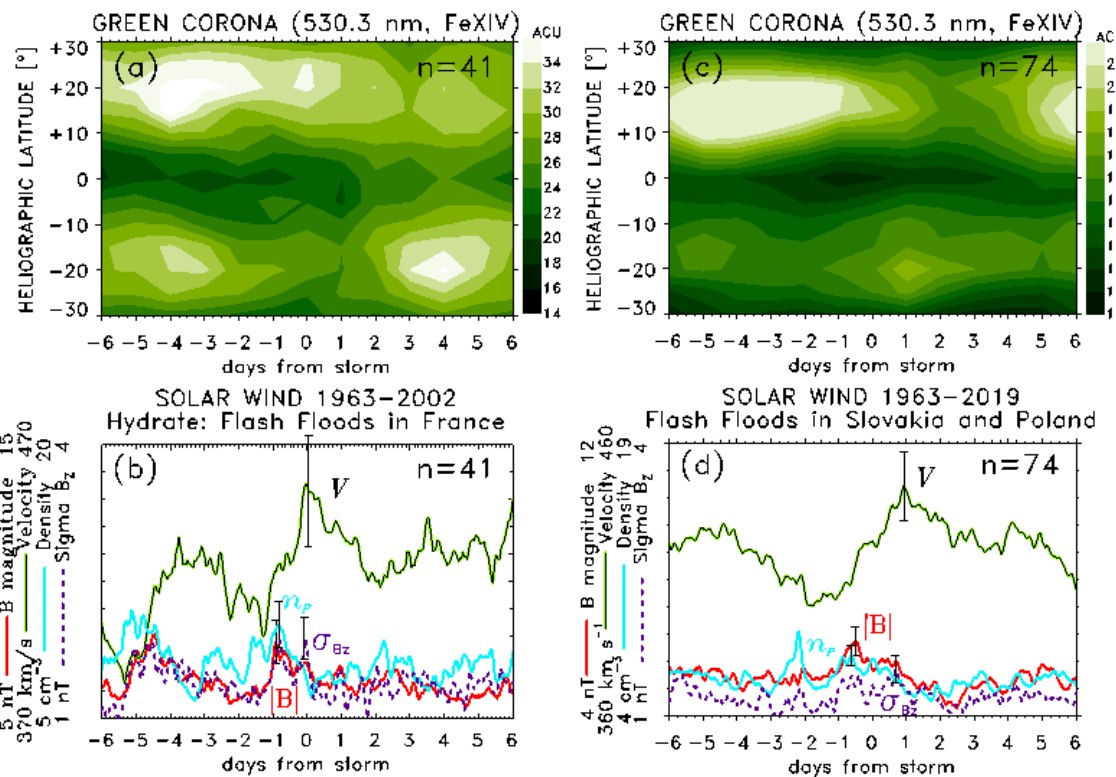

**Figure 1.** The SPE analysis of time series of **(a, c)** green corona intensity and **(b, d)** solar wind plasma parameters keyed to
dates of flash floods in France and Slovakia, including flash floods in the Tatra Mountains in Slovakia and Poland.





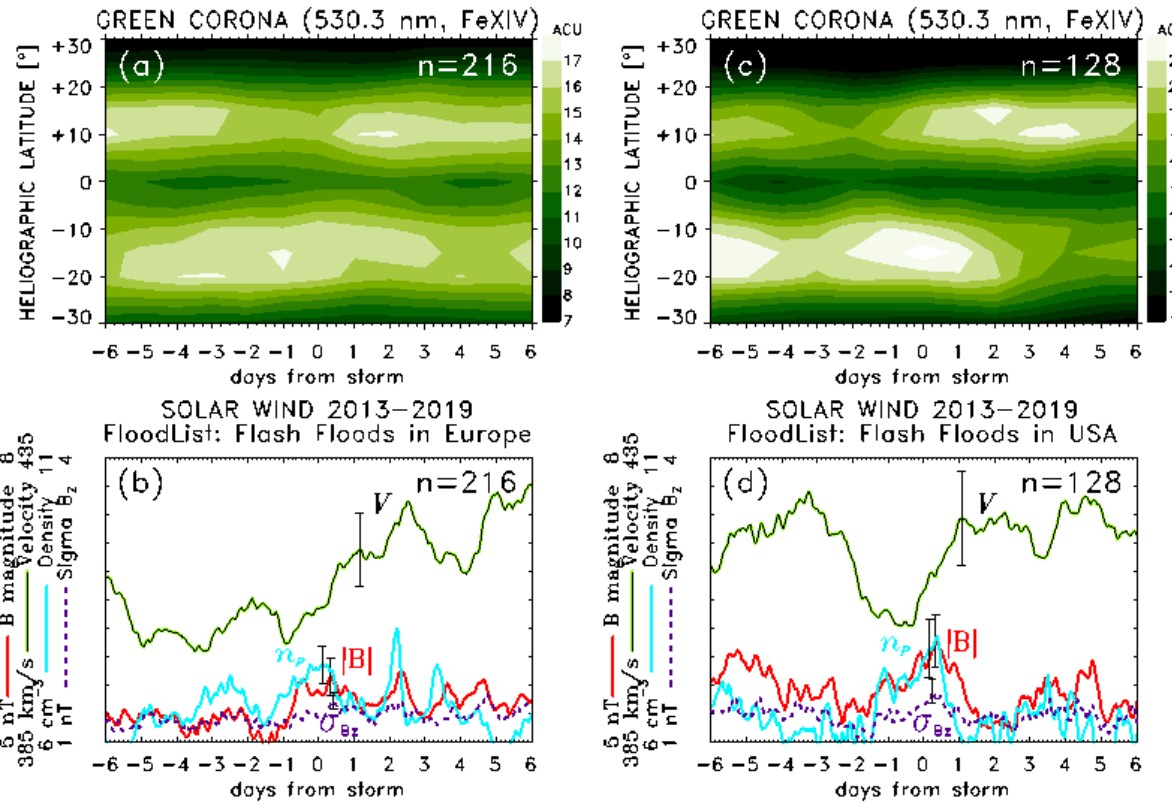

**Figure 2.** The SPE analysis of time series of **(a, c)** green corona intensity and **(b, d)** solar wind plasma parameters keyed to start date/time of heavy rainfall and/or flash floods in Europe and USA.

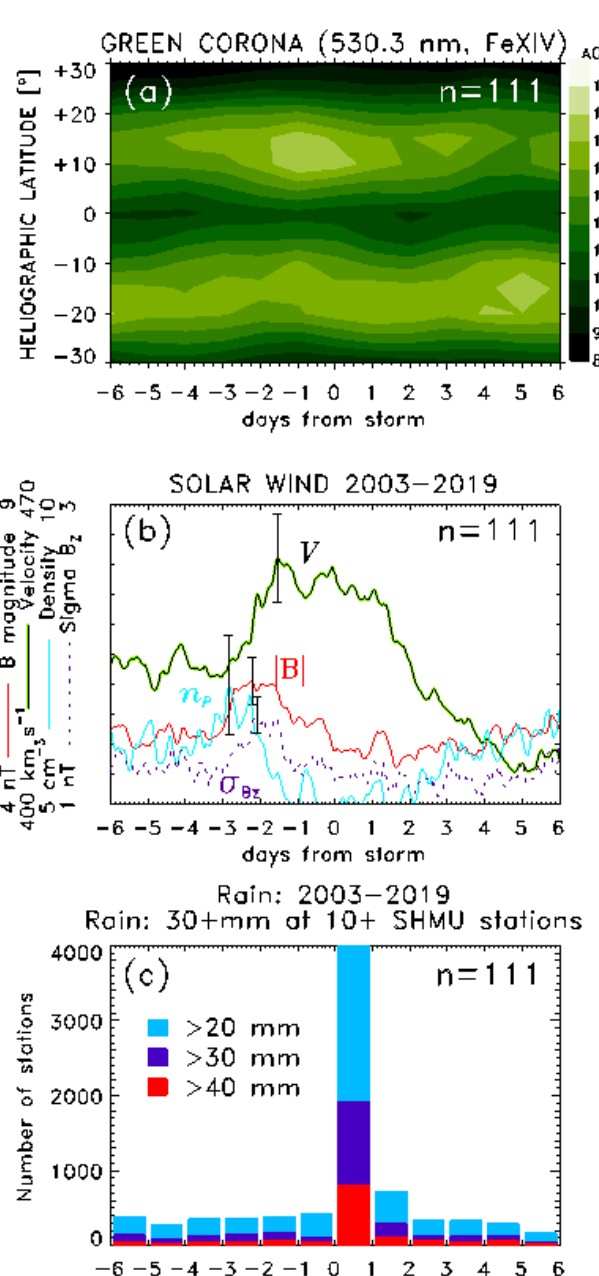

**Figure 3.** The SPE analysis of time series of **(a)** green corona intensity and **(b)** solar wind plasma parameters keyed to start date of heavy rainfall of at least 30 mm/24h recorded at 10 or more stations. **(c)** Total cumulative number of stations that recorded 24-h rainfall exceeding given thresholds in daily precipitation rates.





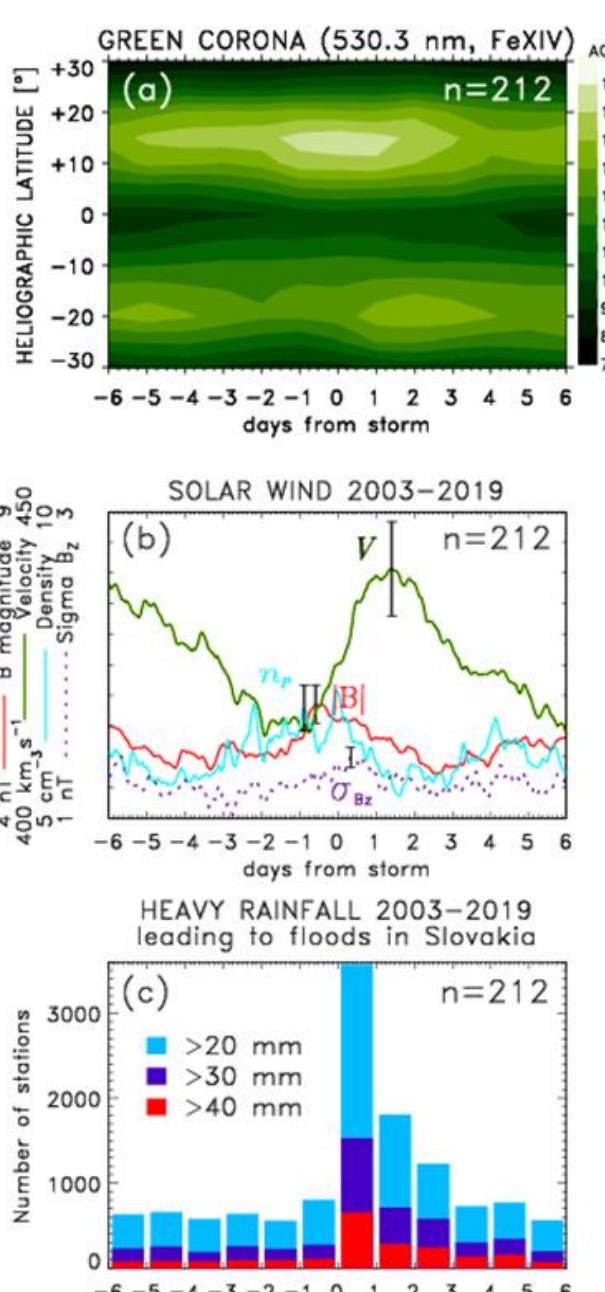

**Figure 4.** The SPE analysis of time series of **(a)** green corona intensity and **(b)** solar wind plasma parameters keyed to start dates/times of significant rainfall leading to floods in Slovakia. **(c)** Total cumulative number of stations that recorded 24-h rainfall exceeding given thresholds in daily precipitation rates.






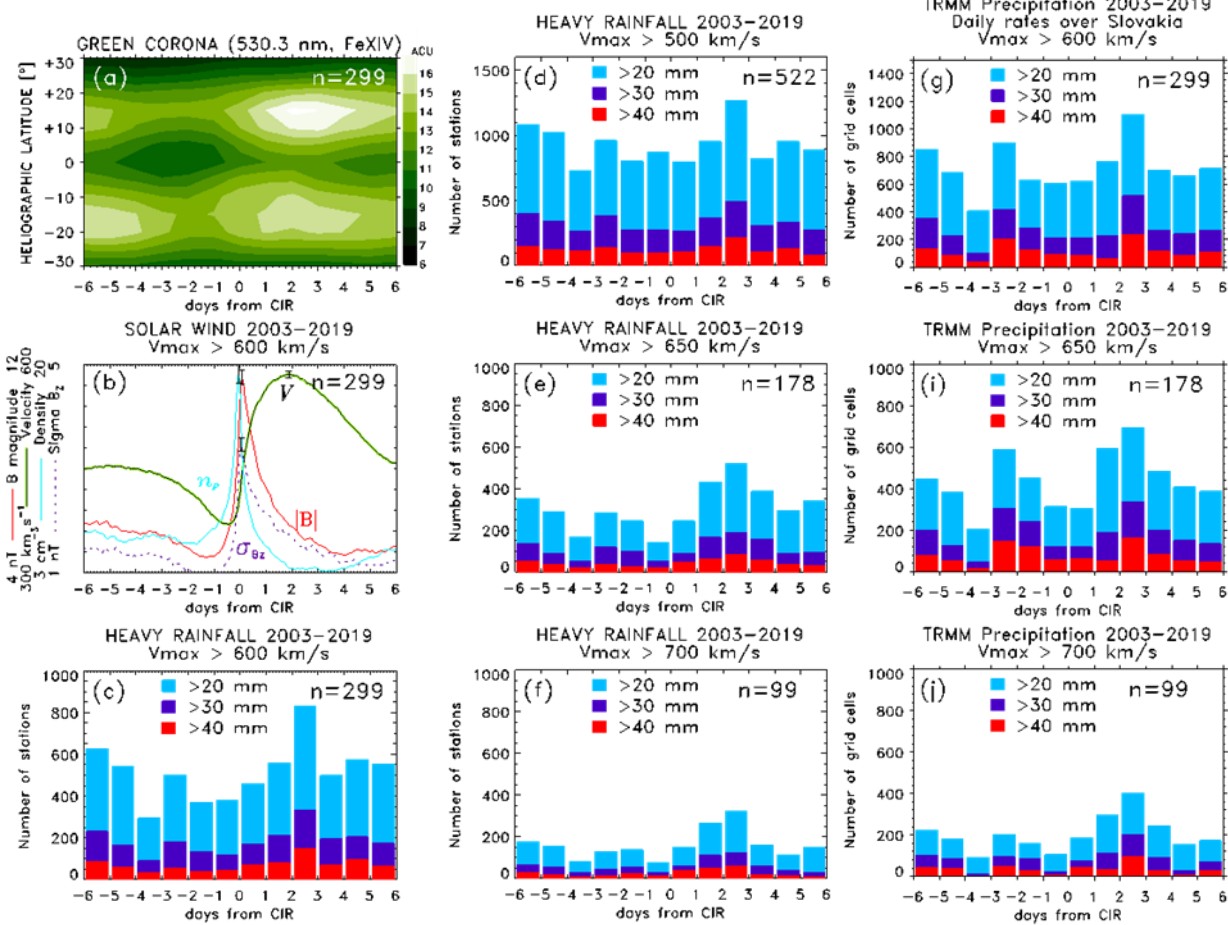

**Figure 5.** The SPE analysis of time series of **(a)** green corona intensity and **(b)** solar wind plasma parameters keyed to the arrivals of HSS/CIRs for solar wind streams that exceeded a given maximum velocity. **(c-f)** Total cumulative number of rain-gauge stations in Slovakia, and **(g-j)** number of TRMM grid cells over Slovakia, with above-threshold daily precipitation rates.





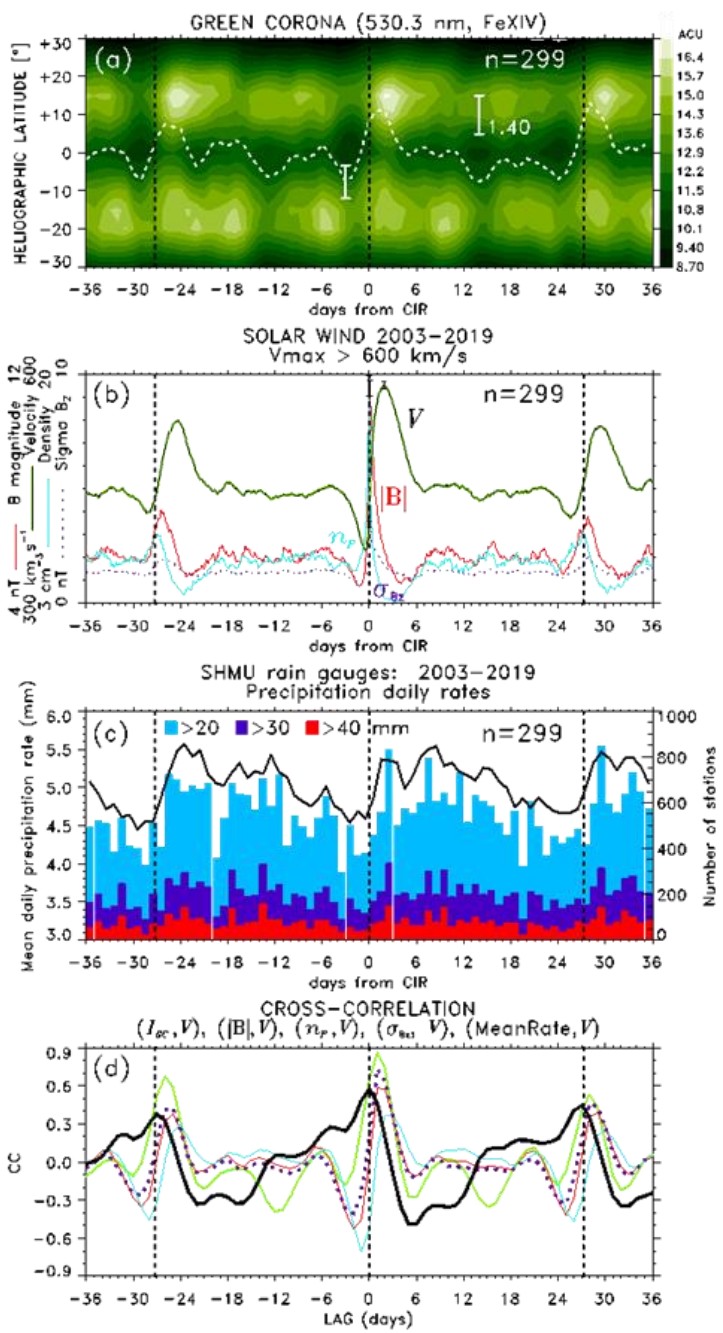

**Figure 6.** The SPE analysis for the period from 2003 to 2019 of **(a)** green corona intensity and **(b)** solar wind plasma parameters keyed to the arrivals of major HSS/CIRs ($V_{max} > 600$ km/s) for periods of ±36 days relative the key time. **(c)** Mean daily precipitation rate and the total cumulative numbers of rain-gauge stations in Slovakia that measured daily precipitation rates exceeding given thresholds. **(d)** The cross-correlation function computed for the solar wind velocity $V$ paired with the rest of the variables, ($I_{GC}$, $V$) in green, ($n_p$, $V$) in light blue, ($B$, $V$) in red, ($\sigma_{Bz}$, $V$) in purple, and (mean rate, $V$) in black.





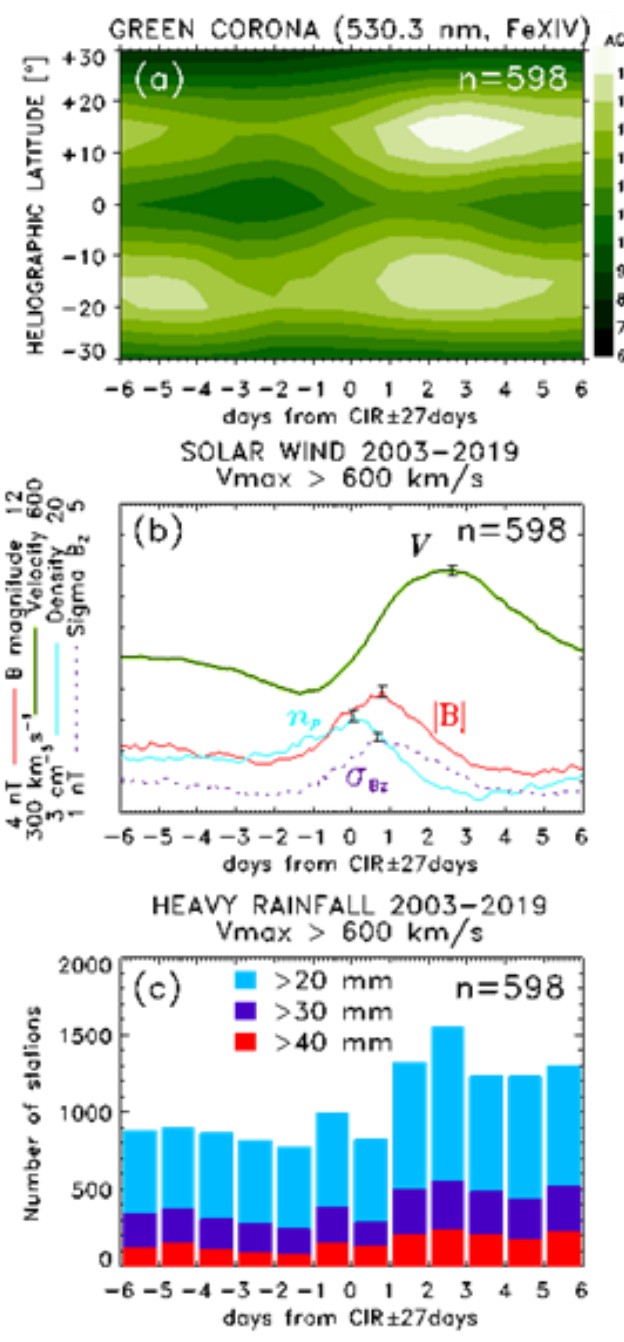

**Figure 7.** The same as Figs. 5a-c except the SPE analysis keyed to times lagged by ±27 days relative to arrivals of HSS/CIRs.







**Figure 8.** Daily averages of (**a**) solar wind velocity and (**b, c, d**) the number of rain-gauge stations in Slovakia that measured daily precipitation rates exceeding given thresholds ordered by 27-day solar rotation. Dashed lines mark approximate arrivals of recurrent solar wind streams.






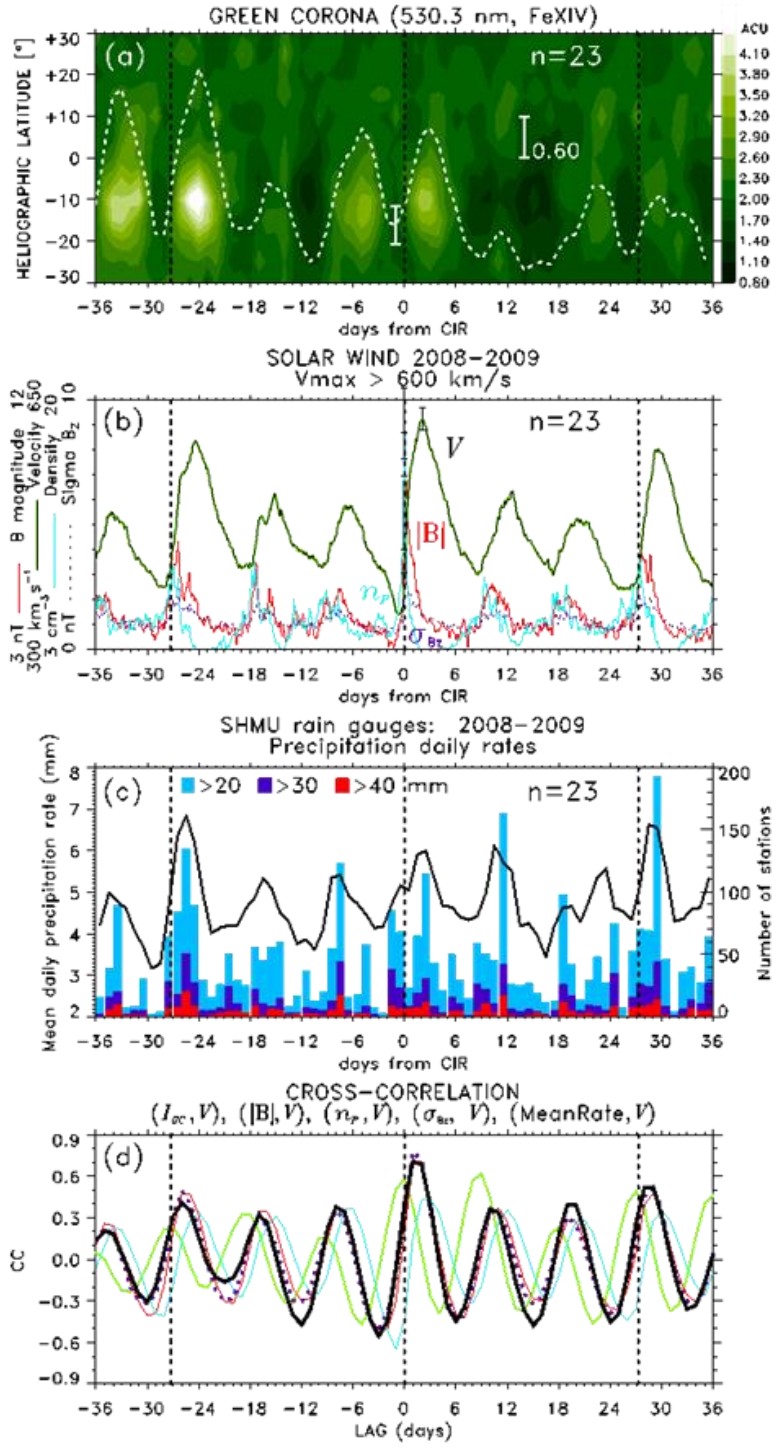

**Figure 9.** The same as Fig. 6 but for years 2008 and 2009.





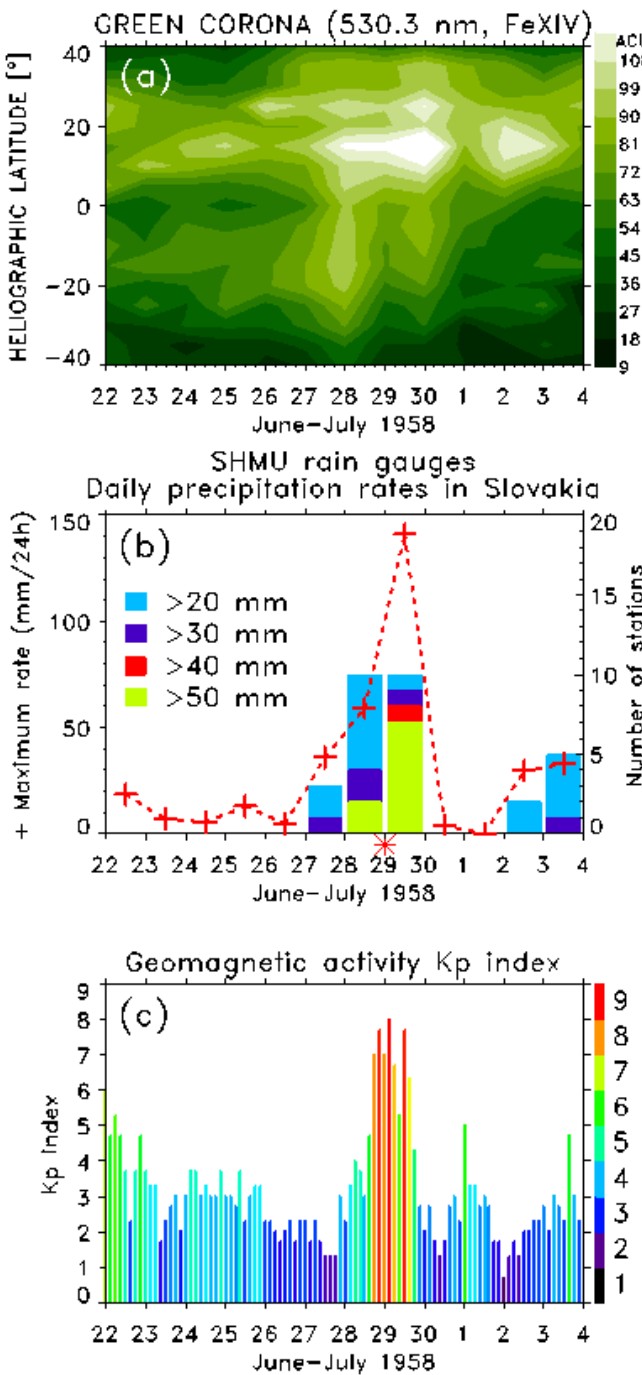

**Figure 10.** Time series of **(a)** green corona intensity, **(b)** maximum daily precipitation rate and cumulative number of stations that recorded daily rainfall exceeding given thresholds, and **(c)** *Kp*-index of geomagnetic activity, for the period that included the flash flood on 28-29 June 1958. The red asterisk marks the interplanetary magnetic sector boundary (Svalgaard, 1975).





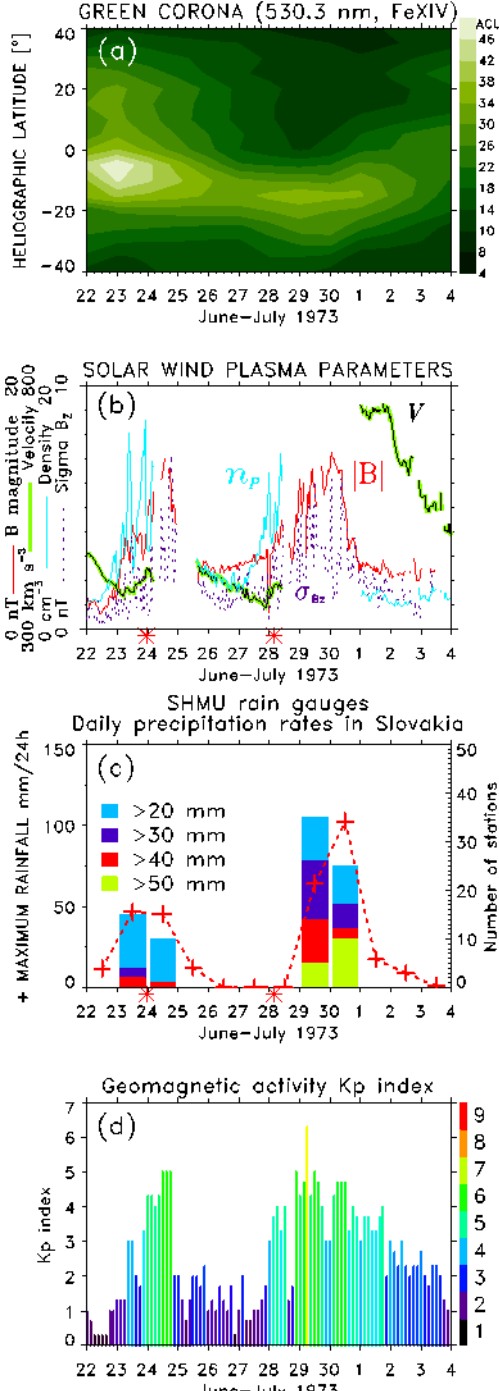

**Figure 11.** Time series of **(a)** green corona intensity, **(b)** solar wind plasma parameters, **(c)** maximum daily precipitation rate

and cumulative number of stations that recorded daily rainfall exceeding given thresholds, and **(d)** *Kp*-index of geomagnetic

activity for the period that included the flash flood on 29-30 June 1973. Red asterisks mark HSS interfaces (CIRs).





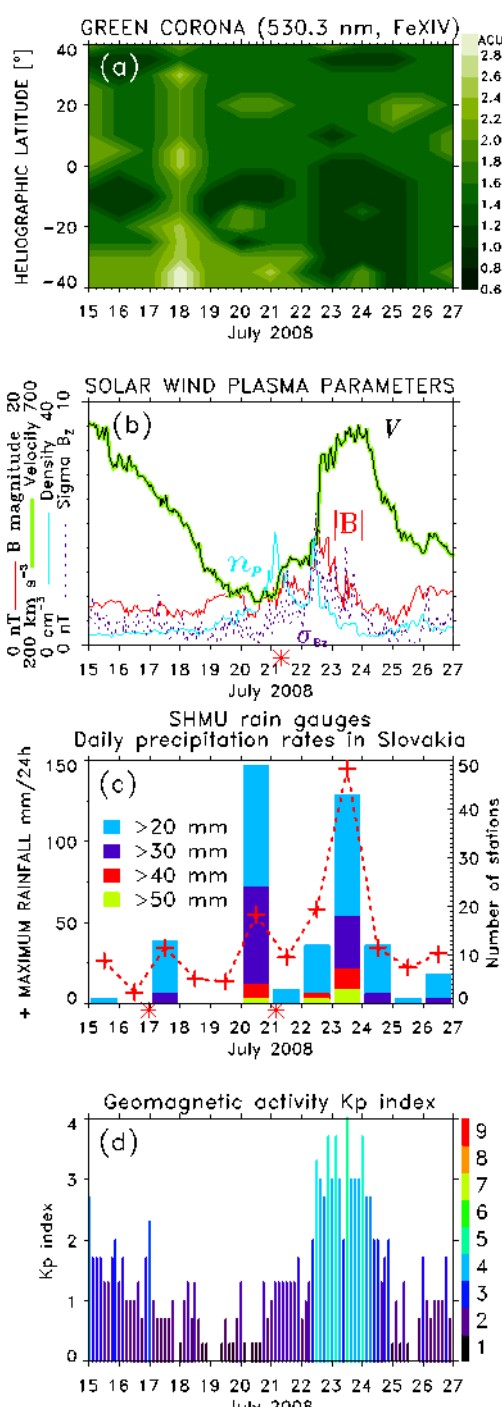

**Figure 12.** The same as Fig. 11 but for the period that included the flash flood on 22-24 July 2008.



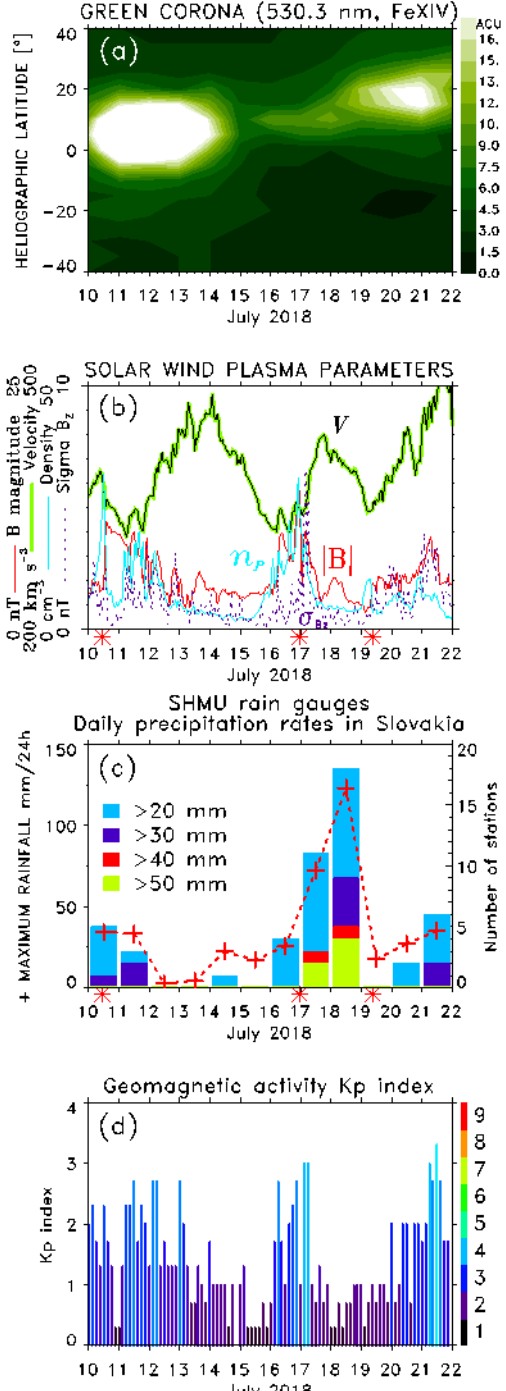


**Figure 13.** The same as Fig. 11 but for the period that included the flash flood on 17-18 July 2018.





**Figure 14.** The OMNI solar wind *V* (solid black line), *B* (red), $n_p$ (broken light blue line) and $\sigma_{Bz}$ (dotted purple line) in **(a)** June 2017 and **(b)** June-July 2018 with the y-axis scales shown on the left. The geomagnetic storm index *Dst* is shown in the middle (green line). Major HSS/CIRs are marked by red asterisks and ICMEs by orange triangles at the time axis. At the top, orange dots show daily rates of rain precipitation at stations in Slovakia and the orange symbols (◊) mark the start days/times of significant rainfall that caused floods in Slovakia. Black symbols mark the start days/times of heavy rainfall/flash floods in (Δ) Europe, (□) USA and Japan (○). In purple, maximum daily rates (solid line) and the number of grid cells with rainfall exceeding 30 mm (dotted line) are from the Deutscher Wetterdienst (DWD) REGNIE dataset.



**Figure 15.** The same as Fig. 11 but for the period that included floods on July 1, 10, and 24, 2017.


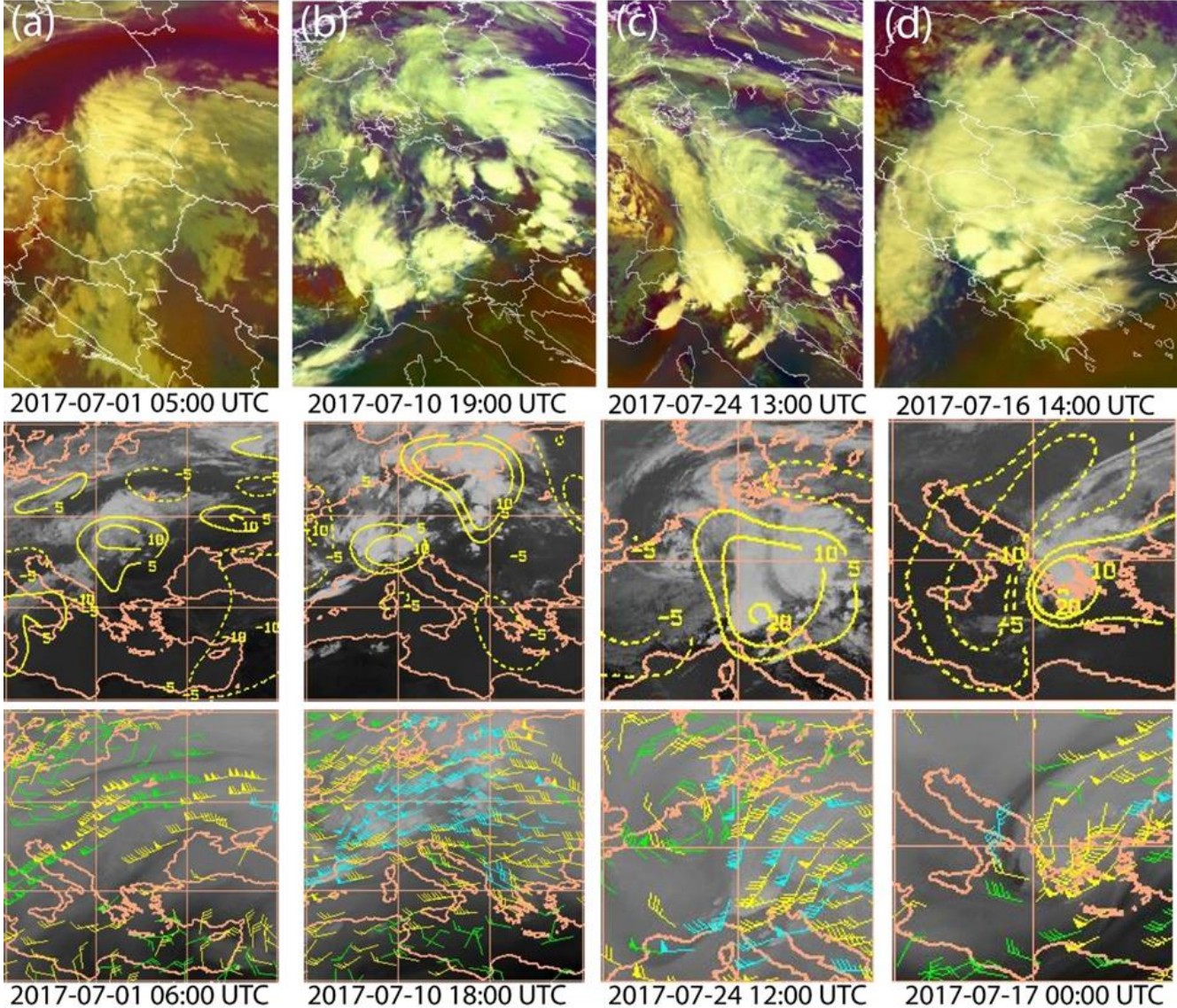

**Figure 16.** The Meteosat RGB Composites Airmass images showing **(a)** a striated delta cloud in the warm frontal zone of an intensifying extratropical cyclone over Slovakia and Poland on July 1, **(b)** a series of convective cells in the frontal zones of intensifying mesoscale systems on July 10, **(c)** vigorous convective activity in frontal zones of mesoscale systems on July 24, and **(d)** a string of convective cells in the intensifying severe weather system Medusa over Greece on July 16, 2017. (https://eumetview.eumetsat.int/static-images/MSG/RGB/AIRMASS/). Overlay data products from the CIMSS data archive of wind analysis (http://tropic.ssec.wisc.edu/archive/) show divergence between 150-300 mb in the middle panels, and mid-upper level winds in the bottom panels, for approximately the same times as those of the images in the top panels.



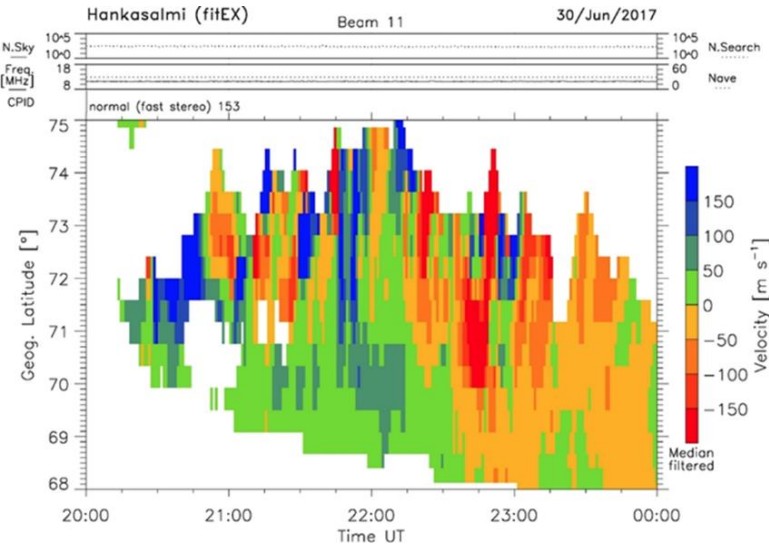

**Figure 17.** The line-of-sight (LoS) velocity observed by the SuperDARN Hankasalmi radar beam 11.

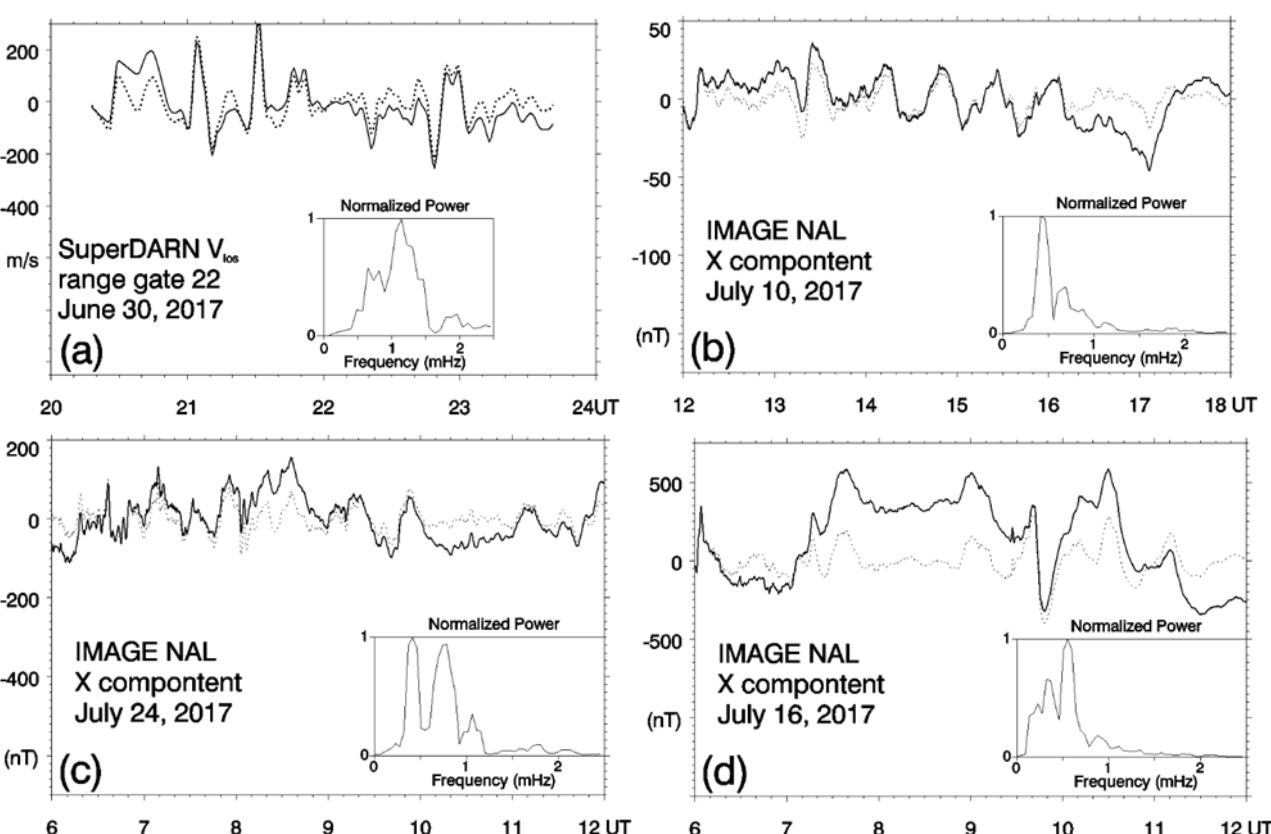

**Figure 18.** The X components of magnetic field and the FFT spectrum of the detrended time series (dotted line) observed by magnetometers in Svalbard on **(a)** June 30/July 1, **(b)** July 10, **(c)** July 24, and **(d)** July 16, 2017.

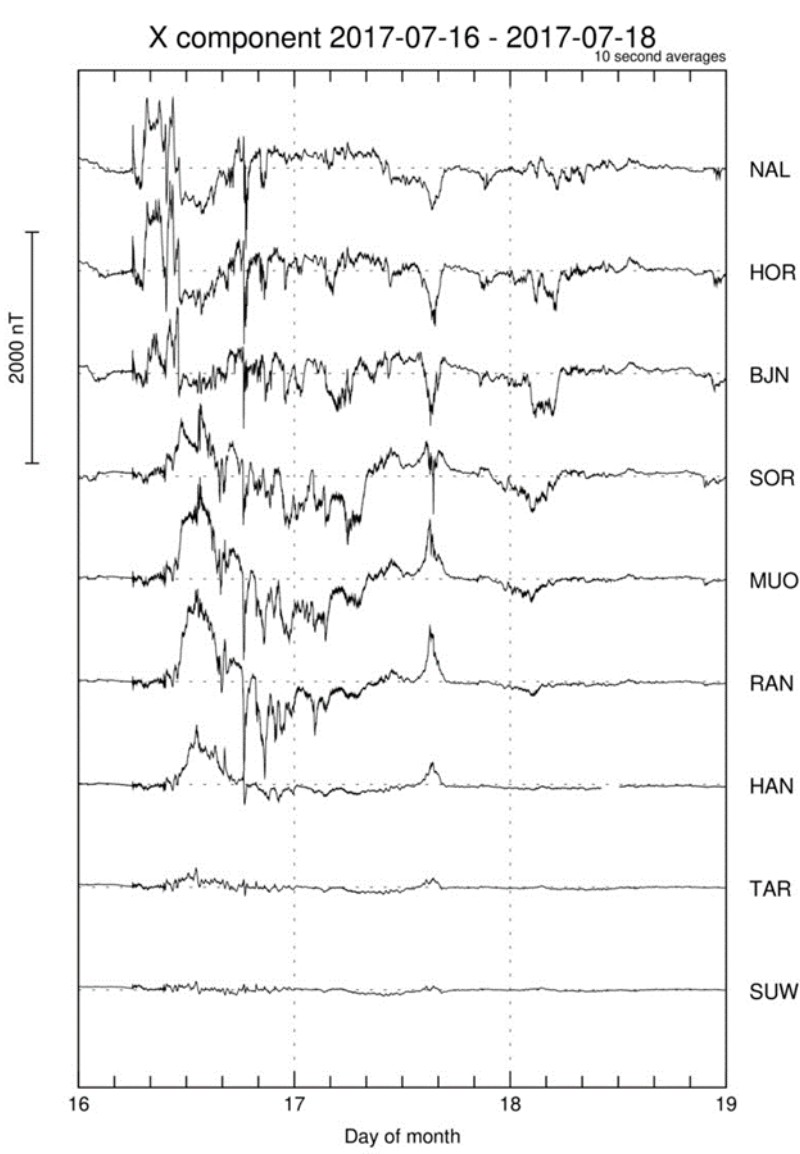

**Figure 19.** The X components of magnetic field observed by the IMAGE array of magnetometers.





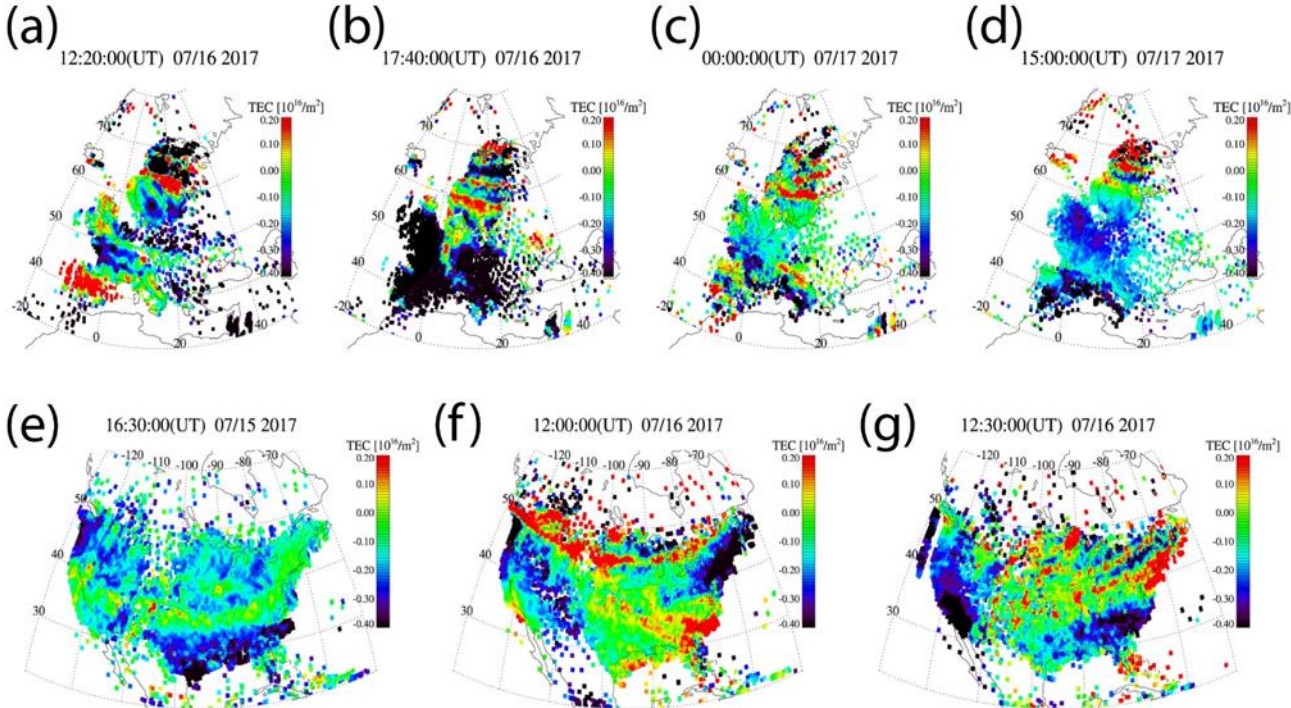


**Figure 20.** Medium- to large-scale TIDs propagating equatorward over Europe and the North America as observed in the detrended TEC.

**Data availability**

All sources of data are described and referenced by URL in Section 2.

**Author contributions**

https://publications.copernicus.org/services/contributor_roles_taxonomy.html

P. Prikryl's roles: conceptualization, formal analysis, investigation, methodology, project administration, resources, software, supervision, validation, visualization, writing original draft, review, and editing.

V. Rušin contributed to acquisition, validation, calibration, and processing of the green corona intensity data, and to preparation of the manuscript (resources; supervision; writing – original draft preparation; writing, review, and editing).

E. A. Prikryl contributed to conceptualization; resources; formal analysis, applicability of statistical, mathematical, computational methods; writing, review, and editing.

P. Šťastný provided mentorship and supervision over the acquisition, validation, calibration, processing and use of the rain-

gauge data, and contributed to writing, review, and editing.

M. Turňa executed the rain-gauge data retrievals, provided required data files and formats needed for the rain-gauge data analysis.





M. Zeleňáková contributed to Annual Flood Reports that included records of significant rainfall events, and contributed to writing, review, and editing.

**Competing interests**

The authors declare they have no conflict of interest.