# Peer review of "Heavy rainfall, floods, and flash floods in the context of solar wind coupling to the magnetosphere-ionosphere-atmosphere system"

_Annales Geophysicae, 2021_

## Author Response (AR1)

We thank the referees for careful reading, and for their comments and suggestions. We are particularly grateful to the second referee providing valuable insights and many helpful references on the topic of solar wind. We have responded to all the comments (see, sections in italics) and made appropriate revisions in the manuscript.

**Reply to Referee #1**

RC1: 'Comment on angeo-2021-23', Anonymous Referee #1, 07 May 2021

The study of P. Prikryl et al. is very interesting. It evaluated a lot of observations and the application of the superposed epoch analysis provided convincing results about a correlation between solar wind perturbations and heavy rainfall. It cannot be expected that the authors can provide a complete physical explanation of this correlation. However, I have two major comments:

1) Please explain why auroral gravity waves with reduced amplitudes at the tropopause (factor 1000-10000, line 512) have an impact on the tropospheric weather. In the troposphere, there are many other perturbations which have larger amplitudes.

**Reply:** This is a commonly raised question about the physical mechanism that we are proposing. The release of conditional symmetric instability, particularly in the warm frontal zone, has been known to initiate convection and result in frontal precipitation bands. We provide references in the revised manuscript. Even an infinitesimally small displacement of a moist air parcel can initiate slantwise convection. In our previous publication we showed that warm frontal cloud bands, sometimes called striated delta clouds, in rapidly intensifying extratropical cyclones, tend to be observed following arrivals of high-speed streams. When down-going AGWs over-reflect in the warm frontal zone of extratropical cyclones, even a small lift they would provide to a moist air parcel that is already rising over the cold air ahead, can initiate slantwise convection, thus forming a precipitation band. Furthermore, the overreflection of gravity waves can result in amplification. Certainly, there are many other sources of gravity waves of larger amplitudes in the troposphere itself, but they would not be synchronized with solar wind events. Also, we are not aware of any reports of a specific type of tropospheric perturbation that would reach the instability region to explain the initiation of convection and its banded structure other than stochastic fluctuations. Although stochastic fluctuations can trigger convection, in our opinion, they would neither explain the observed link to solar wind nor the coherent wave structure of the striated delta clouds. In the revised manuscript we added a paragraph in Section 7.

2) Please add a discussion about alternative explanation, e.g. variation of the ionospheric conductance by TIDs leading to more thunderstorms.

**Reply:** In the Introduction we included a short review of previously observed sun-weather links and alternative physical mechanisms to explain them.

Minor comments:

Fig 10b caption, please explain the red + marks *The symbol is now explained in the caption*.

line 457 Pioneer spacecraft in 1970s was .... (singular?) "spacecraft" is both the singular and the plural form.

**Reply to Referee #2**

RC2: 'Comment on angeo-2021-23', Anonymous Referee #2, 23 May 2021

Review of "Heavy rainfall, floods and flash floods in the context of solar wind coupling to the magnetosphere-ionosphere-atmosphere system" by Prikryl, Rusin, Prikryl, Stastny, Turna and Zelenakova

The paper contains interesting observations of weather related to solar wind conditions, but they have nothing to with sector boundary/heliospheric current sheet crossings. The body of the paper shows that extreme weather is associated with HSSs and that part of the paper should be emphasized. It seems that the authors do not realize that Alfven waves in HSSs cause substantial energy input into the Earth's auroral zones. The physics of the coupling between the solar wind-magnetosphere-ionosphere during HSSs is not discussed or at least not discussed correctly. There has been a research book containing tens of articles and tens of articles separately on this topic which the authors are apparently unaware of. The discussion of the intensity of high speed streams is a bit incorrect.

The paper needs a major rewrite to address the above problems. I will try to indicate where and how and give some references for the authors to follow. The authors need to do a bit of reading before attempting to rewrite the paper.

**Reply:** We are aware of Alfvén waves in HSSs causing substantial energy input into the Earth's auroral zones, but we agree that we have not sufficiently and properly discussed the physics of the coupling. Since we have done it in the previous publications we thought it would lead to lot of repetition. But it is needed here, at least to some extent, we agree. We have revised and heeded very helpful advice and comments by the referee. We emphasise the role of HSSs, but keep and expand on the discussion of HSC, because we need to relate our results to those published by Wilcox in 70's. Also, as the reviewer pointed out, it is the high-density plasma sheath (HSP) adjacent to HCS, or even collocated with HCS, that is important.

**Major Comments**

Almost all of the solar, interplanetary and magnetospheric results presented in this paper are not new. These previous findings should be referenced in an introductory paragraph or paragraphs so the reader is not misled when reading through the rest of the text.

**Reply:** We have no intention to present any new solar, interplanetary or magnetospheric results. What is shown is to provide, as the title indicates, the context for the tropospheric precipitation study, which we focus on. Of course, we should reference previous findings and provide relevant information for the reader not necessarily familiar with the topic of solar wind-MIA coupling. In the previous paper on this subject Prikryl et al. (2019) have included extensive referencing from early discoveries to the most recent results, as well as discussions of solar wind coupling processes. Although it is not advisable to keep repeating all of this in subsequent publications, we have now expanded the Introduction by two paragraphs reviewing and referencing previous findings to guide the reader.

The discovery that coronal holes were the source of high speed solar wind streams was made in 1973 and was published in Sol. Phys. 23, 123, 1973. This paper should be quoted

**Reply: Reference included.**

High speed streams have a more or less constant speed of ~750 to 800 km/s. The authors seem to be unaware of this fact. See Science, 268, 1030, 1995; JGR, 111, A07S01, 2006. doi:10,1029/2005JA011273; JGRSP, 123, 2018. https://doi.org/10.1002/2017JA024203.

**Reply:** We agree that we should be paying more attention to overall structure of solar wind. The reviewer is referring to solar wind at high heliographic latitudes from polar coronal holes, which was indeed observed and documented by Ulysses orbiting the Sun at high inclinations from the ecliptic, as the Science (1995) paper reported. In the ecliptic plane, the Earth is usually not impacted by HSS from polar coronal holes unless they extend to low latitudes. Such regions of extended polar coronal holes are sources of high-speed solar wind at the edges of these HSSs with speed ranging from ~400 to 800 km/s as observed, for example, by ACE near the Lagrangian L1 point. The second paper provides a relevant review of HSS/CIR and it is now referenced in the revised manuscript. The third paper, another review paper by Tsurutani et al. (2018), has Figure 22 taken from Hellinger et al. (2013) that shows the above range of solar wind speeds.

So why do HSSs detected at 1 AU sometimes have peak speeds less than 750 to 800 km/s? The solar wind emanating from the center of the CH has the speed of 750 to 800 km/s. However the solar wind expands at greater than 1/r2 which is called "super radial expansion". Detection of the HSS at Earth does not always cross the center or main body of the CH, thus the speed can be less than this maximum. This is shown and discussed in pp.99-119 in "Midlatitude Ionospheric Dynamics and Disturbances", vol 181 AGU press, 2008 (fig 3). 10.1029/181GM11; JASTP 73, 164, 2011 doi:10.1016/j.jastp.2010.04.003 and AG, 29, 839-849, 2011 doi:10.5194/angeo-29-839-.2011 (see schematic). Thus the profile of your HSS events have a rise to their maximum speed and then a tapering down. If the HSS event does not reach a peak speed of 750 to 800 km/s, then the spacecraft is just catching the edges of the superradial expanded part of the HSS. The HSS does not have variable speeds as your writeup seems to indicate. Please correct this throughout the paper.

**Reply:** Continuing our reply above, we agree that a spacecraft in the ecliptic plane can only catch the edges of this HSS when the polar coronal holes extend to low latitudes. But coronal holes also form in regions centered on the open magnetic field flux in between the active regions of closed magnetic field at low latitudes. As a result, the Earth is repeatedly immersed in the slow- and high-speed solar wind of a variable speed, as it is crossing the warped heliospheric current sheet (HCS) separating the opposite polarities of the interplanetary magnetic field. This is what is relevant for the solar wind-MIA coupling. But the point made by the reviewer is well taken and statements of the maximum speed are corrected. In short, in the manuscript we refer to the maximum speed **observed** near the Earth, not the overall maximum of the entire HSS emanating from somewhere near the center of the coronal hole, polar or low latitude one. We have now clarified this in the revised manuscript.

There are many different interplanetary high plasma density features besides CIRs that can be found adjacent to CIRs. In your superposed epoch you are detecting all of them. For a schematic of these features, please see JGR, 100, 21717, 1995 and the paper starting on p 1 in Recurrent Magnetic Storms: Corotating Solar wind Streams, AGU monograph 167, 2006. One particular feature is the high density heliospheric plasma sheet (HPS) which will be mentioned again later.

**Reply:** We are aware of the important regions adjacent to CIRs, including HCS/HPS, and we have referenced these in a previous publication (Prikryl et al., 2019). They are now mentioned in the introduction paragraph.**

CIRs were discovered in GRL, 3, 3, 137-141, 1976. This should be cited in the Introduction Section. CIRs are compressive regions caused by the fast wind interacting with upstream slow solar wind and therefore have magnetic field strengths higher than in the quiet solar wind. They have highly varying Bz components which can cause magnetic reconnection (see and quote PRL, 6, 47, 1961 for the solar wind energy coupling mechanism: magnetic reconnection and the JGR 1995 paper and references for Bz variations). However because of the strongly varying Bz within the CIR, the magnetic storms they cause are not particularly strong (see the above JGR 1995 paper).

**Reply:** As we mentioned above, all these publications have been discussed and referenced in our previous papers. For the reader's convenience we have referenced them again.**

The HSSs which are the main focus of your paper have embedded interplanetary Alfven waves within them. Through magnetic reconnection associated with the southward component waves with northward dayside magnetopause magnetic fields, strong auroral zone activity is caused. This was first reported in PSS, 35, 405-412, 1987. The resultant geomagnetic activity is called HILDCAAs, an acronym that is commonly used now in the literature. An entire AGU special issue was devoted to HSSs: "Recurrent Magnetic Storms: Corotating Solar Wind Streams", vol 167, AGU press, 2006. 10.1029/167GM03 following an AGU Chapman meeting in 2005. This book should be quoted. It was first pointed out by the JGR 1995 paper that HSSs put more energy into the magnetosphere over 4 days than a ICME storm does. This feature of HSSs/HILDCAAs was verified by articles within the 2006 AGU book: https://doi.org/10.1029/167GM24, and https://doi.org/10.1029/167GM11. THIS is the energy input into the magnetosphere and not the sector boundary/heliospheric current sheet. Or probably CIR magnetic storms.

**Reply:** Thank you for pointing this out. Of course, we are aware of Alfvén waves and should have included more discussion of it also in the present manuscript. This is now corrected in the revised version. Regarding SBC/HCS, we have not linked it to the energy input into the magnetosphere. As already mentioned in Wilcox et al. papers, and we have also discussed it earlier (Prikryl et al., 2009), these boundaries are just convenient markers often indicating an imminent arrival of HSS 1-2 days later, sometimes SBC/HCS coincide with the stream interface. This is the reason why we think it is important to discuss the Wilcox effect and its relation or similarity to the present results. More on this below.

HSSs detected at Earth during the solar declining phase and solar minimum are well known. Please cite JGR 1995, JASTP, 121, 24-31, 2014, JGRSP, 119, 2675-2690, 2014. Doi:10.1002/2013JA019646.

**Reply: Will do.**

Another physical mechanism has been proposed to replace any sector boundary/heliospheric current sheet magnetospheric effects. This is the heliospheric plasma sheet (HPS). This is the high density plasma located in the slow solar wind just ahead of the CIR and collocated with the HCS (see the schematics in the 1995 and 2006 papers). The ram pressure compressing the magnetosphere from HPS impingement cause the precipitation of relativistic electrons creating heating in the thermosphere, perhaps launching gravity waves. This data proof and scenario is given in JGRSP, 121,

doi:10.1002/2016JA022499. This alternate scenario should be mentioned whereever the Wilcox et al 1973 scenario is mentioned. This mechanism is far more concrete than the sector boundary crossing scenario. The deposition energy of the relativistic electrons is actually calculated whereas the HCS crossings have not been.

**Reply:** We agree. These and a more recent paper are cited and discussed in the revised manuscript. We have already addressed the reviewer's concern that we are relating HCS to energy input. We are not. But we will still use the sector boundary, calling it now consistently HCS, in the data analysis. Nevertheless, as stated already by the reviewer and by Tsurutani et al. (1995), "high-density heliospheric current sheet (HCS) plasma sheet, which leads to field compression" and "The initial phases of these corotating stream-related storms are caused by the increased ram pressure associated with the HCS plasma sheet and the further density enhancement from the stream-stream compression". So there is relation between HCS and HPS.

Paragraph starting on line 36. The other mechanism for the Wilcox et al 1973, 1974 tropospheric vorticity observations should be mentioned here with equal weight. I will mention later that perhaps BOTH the Wilcox et al and the HPS impingement mechanism discussions should be reduced or deleted? There is a better mechanism which explains energy input during HSSs.

**Reply:** Both "mechanism" are discussed now. Of course, we are emphasizing the energy input during HSSs.

Line 44. "Prikryl et al. (2009a) confirmed the "Wilcox effect"" may be an incorrect statement. The weather may instead be due to CIR storms or HSS/HILDCAA effects. This should be corrected/addressed here. In fact the bulk of the paper has to do with HSSs and therefore the Wilcox effect discussion seems to be superfluous.

**Reply:** To clarify this - we confirmed the **observation** of the "Wilcox effect" using data for the northern and southern hemisphere. We have replaced "confirmed" with "observed" in the revised manuscript. We are not saying that we have confirmed any physical mechanism that might explain it. There have been none accepted by the wide scientific community so far. That is the reason why we have been focusing on HSS Alfvén waves coupling to MIA generating gravity waves that can trigger convection in the troposphere as we have discussed in this and previous papers. The "Wilcox effect" has started it all, and has not been adequately explained yet, so the discussion is not superfluous.

Line 50. Here you say that flash floods follow HSSs? Which is it, SBs/HCSs, heliospheric plasma sheet impingements, CIR storms or HSSs? HPS impingements were hypothesized to cause AGWs as well. This paragraph needs to be modified. Again the paper focusses on HSSs and NOT on the HCS crossings.

**Reply:** We have already addressed this question earlier. We have cited Trurutani et al. (2016), and yes, we agree that the focus is on HSSs but define the key times by CIRs at the leading edge of HSSs.

Line 73. Polar coronal holes. A good reference is GRL, 21, 1105-1108, 1994. https://doi.org/10.1029/94GL01065. "Can extend to low heliographic latitudes" see the JGR 1995 paper. A good reference here.

Reply: Thank you. We will cite more papers.

Line 109. "ICMEs" needs a reference. I suggest JGR, 86, A8, 6673-6684, 1981 for the magnetic cloud and JGR, 93, A8, 8519-8533, 1988 for "driver gas" and upstream shock/sheath. A more correct phrase would be "ICMEs and their upstream sheaths". Sheaths have been shown to be equally as effective as magnetic clouds for causing magnetic storms: JGRSP, 124, 3926-3948, 2019. https://doi.org/10.1029/2018JA026425 and references therein.

**Reply:** Thank you for suggestions. Although the ICMEs are not in the focus in this manuscript their impacts certainly generate large-amplitude gravity waves.

Line 116 to 120. This is all related to the HSS and has nothing to do with the Wilcox effect. Delete reference to the Wilcox et al effect throughout the paper?

Line 134. Again this is all related to the HSS/HILDCAA. Delete Wilcox effect reference?

**Reply:** Although we agree that the physical cause of the observed relationship is related to HSS/HILDCAA, we retain and expand the reference to, and discussion of, the Wilcox effect. Distributions of HCSs are now plotted in Figures 6 and 9 to show what has been already stated in the Introduction, i.e., HCSs precede CIRs by ~1 day. If the SPE analysis is keyed to HCSs instead of CIRs, the patterns of an increase in high-rate precipitation from minima to maxima are shifted by about one day thus making the results similar to those observed by Wilcox et al. for VAI. Please, note again that we are not relating the observed relationship to SBC/HCS energy wise, just comparing the Wilcox effect with the present results.

Line 159-160. Please see previous discussion of HSSs and CHs. For this case the spacecraft is not sensing the HSS coming from the center of the CH. It is catching the edge of the HSS.

**Reply:** We have already commented on this earlier. It is now clarified that we are referring to the observed maximum in the solar wind speed as measured at Earth, not the overall maximum of the large-scale HSS from a polar coronal hole.

Line 161-162. Be careful here. You are probably detecting the HPS as well.

**Reply:** Key times on Figs. 5a-c are defined by the stream interfaces (CIRs). So the HPS, and HCS, if present, would precede it.

Line 173. Polar coronal holes and their HSSs have been known to persist for years. Please see and quote JGR 87, A9, 7389-7404, 1982 and JGR, 100, A11, 21717-21733, 1995 here. This interval 1973-1975 was a record interval for HSSs impacting the Earth's magnetosphere. Such an occurrence has not happened again. After this paper, you might want to look at this period for extreme weather?

**Reply:** Thank you for references and suggestion. Maybe one day we may try examine it, although at the moment we do not have any data on severe weather for those years.

Line 194. A recurrence period of 27 days is not a new finding. See previous references.

**Reply:** Not for HSSs, but new for weather.

Lines 204 to 206. HSSs have their sources at CHs, but CIRs are due to the HSSs interacting with slow solar wind plasmas in the interplanetary medium. This has been modeled by the paper starting on p. 51, in Coll. Shocks Helio: Rev. Curr. Res. AGU press, vol 35, Wash DC, 1985. The 7 to 9 day period may not

have to do with CHs being separated on the Sun. It is also possible that you may be detecting the wavy edges of a single polar coronal hole. So be careful in your statement.

**Reply:** In 2008-2009, HSSs were caused by approximately circular and compact midlatitude coronal holes (De Toma, 2011), not necessarily the "wavy edges of a single polar coronal hole"

Line 208. One should add the reference AG, 29, 839-849, 2011 here. Please note that during solar minimum an all time minimum in geomagnetic activity occurred due to the low HSS velocities at Earth. An interesting question for you is did you see a minimum in extreme weather at this time? Maybe another paper?

**Reply:** The reference is added. The 2008 recurrent CHs were indeed recurring very regularly. Not so regular in 1996/97 (Fig. 6 in the 2011 paper), and neither in 2018. Something to examine in the future.

Lines 211-214. This is somewhat of a repeat of the above paper?

**Reply:** Sure, it is the same recurrent CH/HSS causing recurrent geomagnetic activity. We have included it elsewhere (Prikryl et al., 2012, their Fig. 7) to show recurrent GPS scintillation activity. Here we go again suggesting a recurrent high-rate precipitation in the troposphere, something is new.

Line 233. I disagree that "the results are similar to the "Wilcox effect". These are HSS or CIR storm effects which occur much later in time. Please revise this discussion.

**Reply:** Not much later. We have updated Figs. 6 and 9 to show cumulative numbers of HCS crossings for each epoch day. They all peak about one day before CIRs. So if the SPE analysis is keyed by HCSs instead of CIRs, the patterns shift forward ~1 day, which is similar to the "Wilcox effect". The shift is indeed because the HSS or CIR storm effects occur later. Also, as mentioned by Prikryl et al. (2009a), the minima in the VAI and high-rate precipitation are likely related to the "calm before the storm in CIR/interactions" (Borovsky and Steinberg, 2006). The discussion on the similarity to the "Wilcox effect" in the last paragraph in the revised Section 4 is amended.

Line 272. 750 km/s is NOT an "exceptionally fast HSS" as discussed before. Please revise discussion associated with Figure 11.

**Reply:** Ok, we revised to "very high". But it is still unusually high in the ecliptic plane. Surely there are ICMEs with higher maximum speeds, but not so many HSSs. In one of our previous papers dealing with tropical cyclones (Prikryl et. 2019) we presented a case of a rapid intensification of the only one ever observed hurricane Catarina in the southern Atlantic following arrival of an HSS with maximum speed exceeding 800 km/s. One of the reviewers, renown as expert in solar wind studies, was surprised to see such high-speed HSS. He would call it exceptional.

Lines 274-275. "moderate geomagnetic storms". Yes that is all CIRs cause. Only ICMEs or their upstream sheaths trigger superintense storms. See previous references.

Reply: Yes, we agree.

Line 281. I would not call 650 km/s a "strong HSS".

**Reply:** Revised to "relatively fast" HSS.

Line 289. This is most likely associated with a HILDCAA.

**Reply:** We agree. Large amplitude ULF waves are often caused by large-amplitude solar wind Alfvén waves.

Line 324. "a very strong and structured HSS"??

**Reply:** There were multiple maxima during this more than a week-long HSS caused by a large and structured CH. https://www.spaceweather.com/archive.php?view=1&day=19&month=07&year=2017

Line 410. HSSs generally do not cause geomagnetic storms. See the 2006 paper. Please double check.

Reply: We agree. But this was a combined effect of HSS/CIR and ICME.

Line 427. "While the HSS was weakening". This is almost certainly a spatial efect rather than a temporal effect. The Sun is rotating and the CH rigidly rotates with it (see the 1982 paper for rigid rotation). So you are detecting the superradial expansion part of the HSS.

**Reply:** We agree. It is the observed velocity that was decreasing as the Earth was exiting the HSS. Revised to: "As the Earth was exiting the HSS".

Lines 452-455. Here you need to mention HILDCAAs and auroral zone auroral activity associated with HSSs. This is what is causing your weather effects and not sector boundaries/HCSs, HPSs or CIR magnetic storms in my opinion. Thus the discussion and conclusions of the paper needs to be revised substantially.

**Reply:** We absolutely agree. Isn't it what we are saying in this paragraph? HSSs dominated by Alfvén waves. These are HILDCAAs, as referenced (Tsurutani and Gonzalez, 1987). In the revision we included the acronym. When we mention SBCs/HCSs or CIRs, we are just saying that there is a tendency of severe weather events to follow these "markers" in the solar wind, a day or a few days later. We think there is no need "to be revised substantially", just to make this point clear. We are aware of Alfvén waves in HSSs causing substantial energy input into the Earth's auroral zones. However, no matter how substantial it may be, by itself it is likely not going to cause any effect on weather, because it is deposited in the upper atmosphere. So saying that "HILDCAAs cause the weather effect" is not quite right either. If anything, it is an indirect effect mediated by AGWs, which we have proposed. And large-amplitude gravity waves are generated already with the arrival of CIRs, not only by HILDCAAs in the body of the HSS. And while we are here discussing statistical occurrence of high-rate precipitation, some of the presented cases of heavy rain and floods occurred coincident with CIR or just after CIR, or even HCS (Fig. 10).

---

## Author Response (AR2)

*Our responses to Referee #2 are shown in italics.*

Review of "Heavy rainfall, floods, and flash floods in the context of solar wind coupling in the magnetosphere-ionosphere-atmosphere system: by Prikryl, Rusin, Prikryl, Stastny, Turna and Zelenakova

The paper contains interesting and new information on solar wind coupling to the Earth's atmosphere. Although the authors have written other papers on this topic prior to this one, the broad range and thorough coverage of the present paper is the best yet and I suspect that this one will become a classic. I have only a few minor comments.

Minor Comments

Title: The title does not say specifically what the "solar wind feature" exactly is. I suggest that you change the title to "Heavy rainfall, floods and flash floods due to high speed solar wind (HSS) coupling to the …..". The solar, solar wind and magnetospheric science community have basically ignored the Wilcox effect (even though Wilcox was a solar/interplanetary scientist) because it was obvious that there was no substantial energy transfer to the Earth due to sector boundary crossings. A reader of the present title might think "ho hum, probably another HCS crossing paper" and not read it. The connection of weather to high speed streams/HILDCAAs is far more plausible and should attract far more interest from the AG readership. I therefore suggest a modification of the present title.

*Reply: We agree and changed the title: "Heavy rainfall, floods, and flash floods influenced by high-speed solar wind coupling to the magnetosphere-ionosphere-atmosphere system"*

Lines 17 and 19, the "solar rotation period". In perhaps the first line where you mention a 27 day periodicity, you could mention that that is the "solar rotation period"? There might be some people that might not know this.

*Reply: Done*

Now for the people that know solar physics, it is well known that the Sun is a differential rotator. ~27 days is the rotation period of surface features at the solar equator as view from the Earth. The rotation period at higher solar latitudes is larger, reaching ~31 days near the poles. Coronal holes are "rigid rotators" rotating at the equatorial rate (even though the coronal holes are not at the equator). None of this needs to be gone into here, but you should mention it somewhere within the text. I will point out one place to do it later.

Line 68. It would be helpful to the readership to let them know that the occurrence of HILDCAAs during HSSs indicate substantial energy input into the auroral zone ionosphere. This substantial energy input differentiates this mechanism from that of HCS crossings (without making a judgement as to which is a better mechanism). About the auroral zone, later in the paper you allude to auroral zone effects without tying it to HSSs and HILDCAAs. So in that line add "indicating substantial energy input into the auroral zone ionosphere"?

*Reply: Done*

Line 78. … high speed solar wind streams?

*Reply: Corrected – adding "streams"*

Line 84. Can you give references to "moist instabilities"? That would help the readership.

*Reply: References added.*

Lines 95-96. The scale size of HSSs is much larger than the magnetosphere or the distance from Earth to the L1 libration point where the satellite is orbiting. So it is expected that what is detected at L1 is the same wind that impacts the Earth's magnetosphere. So perhaps this sentence should be deleted?

*Reply: Deleted*

Line 105. Polar coronal holes are most prominent during solar minimum. So perhaps a more correct sentence would be "…most prominent during solar minimum in polar regions…." ?

*Reply: Agree. Corrected.*

Line 205. Perhaps you can put in the rigid rotators here? "Polar coronal holes and the HSSs have been known to persist for years and appear as "rigged rotators" at the Sun's equatorial rotation period (27 days as observed from Earth)" The Pioneer observations of CIRs were noted to occur at ~25 day periods because it was not viewing the Sun from a rotating viewpoint.

*Reply: Sentence is now amended: "Polar coronal holes and their HSSs have been known to persist for years (Tsurutani et al., 1982; Tsurutani et al., 1995) and appear as "rigid rotators" at the Sun's equatorial rotation period (27 days as observed from Earth)."*

Line 495. You may wish to again emphasize auroral zone energy deposition? "… known to cause HILDCAAs and substantial auroral zone energy deposition (….)".

*Reply: This is emphasized again in Section 7*

Lines 520-21. I suggest that once the paper has been published, the authors circulate it to weather/atmospheric experts to get comments/criticisms. I am not knowledgeable enough to give you comments on that portion of the work. But this all seems very reasonable to me.

*Reply: Thank you for the suggestion, corrections and all the comments.*